# Diagnostic testing intensity for Legionnaires' disease: Spatio-temporal assessment and its effect on surveillance case reporting, Denmark, 2014–2022

Emmanuel Robesyn[1,2,3,4]*, Søren Anker Uldum[4,5], Karsten Dalsgaard Bjerre[6], Charlotte Kjelsø[2], Marc Struelens[7], Cecilia Stålsby Lundborg[1], Steen Ethelberg[2], Christel Faes[8]

1 Department of Global Public Health, Karolinska Institutet, Stockholm, Sweden, 2 Department of Infection Epidemiology and Prevention, Statens Serum Institut, Copenhagen, Denmark, 3 European Centre for Disease Prevention and Control**, Stockholm, Sweden, 4 ESCMID Study Group for Legionella Infections (ESGLI), Basel, Switzerland, 5 Department of Bacteria, Parasites and Fungi, Statens Serum Institut, Copenhagen, Denmark, 6 Department of Data Integration and Analysis, Statens Serum Institut, Copenhagen, Denmark, 7 Faculty of Medicine, Université Libre de Bruxelles, Brussels, Belgium, 8 Data Science Institute, I-Biostat, Hasselt University, Hasselt, Belgium

* emmanuel.robesyn@ki.se

## Abstract

### Background

The global, national and subnational geographical and temporal distribution of reported Legionnaires' disease is extremely heterogeneous, and it is unknown to what extent this accurately reflects the variation in true disease incidence. We studied how diagnostic testing intensity varied across Danish municipalities between 2014 and 2022, how it influenced epidemiological surveillance, and how testing-adjusted reporting can be used to improve the study of disease incidence and its determinants.

### Methods and findings

We used data from the clinical Danish Microbiology Database and the Epidemiological Surveillance System and considered a tentative causal model of how *Legionella* infections give rise to the observed Legionnaires' disease surveillance data. We fitted spatio-temporal models using an approximate Bayesian inference for latent Gaussian models (INLA), providing probabilistic estimates. These allowed us to identify areas of increased risk and spatio-temporal interaction. Our assessment of the Legionnaires' disease testing intensity in Denmark showed considerable spatio-temporal variation across the country. The estimated municipal annual testing intensity ranged from 128 to 2,446 persons receiving at least one *Legionella* urinary antigen or PCR test per 100,000 inhabitants. The median increased between 2014 and 2022 steadily

which permits unrestricted use, distribution, and reproduction in any medium, provided the original author and source are credited.

**Data availability statement:** Observed counts at province level of tested persons, cases and population are provided in the Tables 1 and 2 of the manuscript and Tables 1 to 4 of the supplementary material S4-S6. In the supplementary material S21-S24 we provide tables with the values of the estimates as shown in Figures 2 and 4 of the manuscript. Demographic and map data, along with computing code, are available at GitHub (https://github.com/erobesyn/dk_testing_reporting) and Zenodo (https://doi.org/10.5281/zenodo.15238532). Additional data from the Danish Microbiology Database (MiBa) and from the Epidemiological Surveillance System of Denmark cannot be shared publicly because they are owned by a third party (Departments of Clinical Microbiology and Statens Serum Institut). Data are available from Statens Serum Institut (contact via serum@ssi.dk, https://en.ssi.dk/about-us/information-about-processing-of-personal-data) for researchers who meet the criteria for access to confidential data.

**Funding:** The author(s) received no specific funding for this work.

**Competing interests:** The authors have declared that no competing interests exist.

from 275 to 620 tested persons per 100,000 inhabitants, reflecting an upward trend. The proportion of tested persons with age over 70 increased from 44.8% to 56.4%. Increasing testing intensity leads to higher case reporting, until testing intensity reaches approximately 1000–1200 tested persons per 100 000 inhabitants. The estimated municipal annual testing-adjusted case reporting ranged from 1.4 to 12.0 per 100 000 inhabitants. The median fluctuated over the study period between 2.5 (range 1.4–6.2) in 2014 and 5.2 (range 1.5–11.7) in 2022 with a flat overall time effect.

## Conclusions

We obtained estimates of the spatio-temporal variation of Legionnaires' disease among Danish municipalities. We quantified the positive effect of testing intensity on Legionnaires' disease reporting and found a threshold of annually testing slightly over 1% of the population above which the yield of new cases does not further increase. Despite limitations and possible bias, our study of testing-adjusted case reporting suggests that no substantial increase in Legionnaires' disease has occurred over the nine-year study period. Instead, case ascertainment by physicians has improved considerably through increased *Legionella* testing, particularly in elderly patients. Insight in the variation of testing intensity and its effect on Legionnaires' disease reporting can be used to improve guidance for Legionnaires' disease diagnosis, to better study determinants of Legionnaires' disease, and ultimately to improve Legionnaires' disease prevention and control.

## Introduction

Since the outbreak in the Bellevue-Stratford Hotel in Philadelphia, 1976 [1], Legionnaires' disease (LD) – or pneumonia caused by *Legionella* – has retained, up to today, the attention of public health authorities. Its causative agent elicits sporadic infections, clusters, and outbreaks on all continents. However, the geographical and temporal distribution of the reported disease is extremely heterogeneous, and it is unknown to what extent this accurately reflects the variation in true disease incidence.

Despite the global occurrence [2], the clinical severity [3], the public health burden (mostly attributable to years of life lost [4,5]), and the treatable and preventable character of this waterborne respiratory disease, many low- and middle-income countries have no effective surveillance in place. Also, in high income countries, including within the EU/EEA, public health authorities register widely varying incidence figures, likely reflecting varying degrees of surveillance quality [6]. This is despite a common EU case definition [7,8] and similar national reporting mechanisms (mandatory notification by physicians and clinical laboratories). In Denmark, the reporting incidence increased from 2.2 per 100,000 inhabitants in 2011–2014 [9], to around 4.2 per 100,000 inhabitants in 2019–2022 [10]. With this, Denmark is in the top three of the EU countries with highest Legionnaires' disease incidence,

along with Slovenia and Italy [11]. As in several other countries [12,13], within-country variation has also been studied in Denmark. In a Danish study, a geographical grid model was built to detect areas of excess disease incidence [14]. Another study assessed Danish surveillance data and concluded that age and sex distribution could not explain variation in provincial reporting incidence [15]. Uldum et al. have put forward the hypothesis that a combination of factors influences the incidence rate of LD in each city, including sequence type and serogroup, colonization rate, and concentration of *Legionella* in water systems [16].

About one per one to two thousand adults per year are thought to suffer from community acquired pneumonia (CAP) [2,17]. It is widely believed that of the CAP caused by *Legionella*, many remain undiagnosed. To account for this underascertainment and estimate the true Legionnaires' disease incidence, etiological studies of CAP aim to provide the proportion of CAP attributable to *Legionella* infection to assign multiplication factors. A systematic review and meta-analysis estimated that around 4.6% of the CAP is caused by *Legionella* [2,18].

As a consequence of under- or misdiagnosis, Legionnaires' disease is often inadequately treated [19]. Furthermore, the lack of etiological diagnosis entails missed opportunities to identify the source of infection and implement preventive action [20]. Although Legionnaires' disease has a number of clinical features that are distinct from other causes of pneumonia [21,22], the diagnosis requires a microbiological test [23,24], as reflected in surveillance case definitions [7] and clinical diagnostic guidelines [25].

The importance of testing intensity to interpret changes in infectious disease reporting has been recognized previously [26–28]. In the Netherlands a study excluded a change in diagnostic intensity as cause of decline in reported Legionnaires' disease cases [29]. Authors of a scoping review, concluded that increased testing for *Legionella* and improved diagnostics may be a driver with moderate impact on increasing incidence [30].

Several studies have also assessed factors related to the diagnostic testing for Legionnaires' disease. The test availability has been highlighted as condition for appropriate diagnosis [31–33]. A Swiss study described an increase in tests at provincial level and found that the guidelines, the diagnostic value of the test, and financial and time considerations, were factors influencing the use of tests [34]. In relation to the timing of testing, Allgaier reported that despite more than two thirds (70%) of positive test results happened between June and October, only about a third (36%) of the tests took place in these months, and called for guidelines to support routine Legionella testing [35].

Furthermore, the type of diagnostic tests used is relevant for the observed disease incidence. In Denmark, routine laboratory diagnosis mainly relies on three methods: the urinary antigen test (UAT), which primarily detects *L. pneumophila* serogroup 1 [24,36,37], PCR targeting *L. pneumophila* [38,39], and broader PCR targeting *Legionella* spp. [24,38]. These methods differ in diagnostic scope and performance and therefore in case ascertainment [24,40,41]. For example, UAT detects primarily *L.pneumophila* SG1, which represented in 2023 in Denmark only 63.4% of culture-confirmed cases [10]. Although Denmark is renowned for its high proportion of bacterial culture confirmation (35.1% of cases in 2023) [10], which is the gold-standard for Legionella typing and source-attribution, culture alone does not contribute to the population testing intensity as it is always combined with other tests.

Diagnostic guidelines describing for which patients to request microbiological Legionella tests do not only matter for individual patient care. They also affect the sensitivity and specificity of surveillance systems based on mandatory notification of clinically diagnosed cases, as for Legionnaires' disease [19].

The other way around, surveillance is relevant for individual medical decision making, as well as for the development of diagnostic guidelines. For doctors to assess the positive and negative predictive value of a test result, they need to have an accurate sense of the disease incidence, for which they rely, knowingly or not, on surveillance data. For drafting clinical guidelines, Woodhead pointed in an editorial back in 2002 at the importance of geographical variation in the pathogen occurrence and other factors. Guidelines may not be relevant to all countries equally [37] and pathogen distribution does not follow national boundaries [17], highlighting the importance of (sub)national incidence estimation. Additionally, together with risk management of pathogenic *Legionella* species in engineered water systems, both the diagnostic guidelines and

surveillance are essential for disease prevention. Indeed, detection of cases is needed to identify the environmental hazards that cause them [42].

Currently, in most countries, public health authorities still have limited access to data about diagnostic testing for infectious diseases. This makes it challenging to measure diagnostic testing intensity comprehensively or estimate it for surveillance purposes, and if used, e.g., during the COVID-19 pandemic, often "changes in testing policies … could not be accounted for and rendered surveillance data poorly comparable" [43–45].

In this study, we intended to assess how diagnostic testing intensity influences the case reporting of Legionnaires' disease, and how data on testing intensity can be used to better learn about the underlying disease incidence. We aimed to provide a justification and methodology for use of comprehensive data on diagnostic testing intensity, or estimates thereof, to better interpret surveillance results and to improve research towards Legionnaires' disease determinants. This in turn is expected to improve prevention and control of the disease.

## Materials and methods

### a)  Scientific model and assumptions

Our estimated quantities and effect are based on a number of causal assumptions, as expressed in a directed acyclic graph (DAG)(Fig 1) [46,47]. We assume that geographical units (municipalities) differ among each other and vary over time in their population size, age- and sex-distribution, and environmental and behavioural determinants of diverse causes of pneumonia, such as Legionnaires' disease and other types of pneumonia. This results in varying levels of, unmeasurable, Legionnaires' disease incidence, which we aim to estimate from surveillance data. In the present study, also the determinants are not measured, except for the demographic characteristics of the areas.

For an individual to be counted as a Legionnaires' disease case, the case definition requires the case to present with pneumonia and have been tested, and the Legionella test to be positive. As a consequence, at the population level, both a change in the disease incidence and a change in the testing intensity level may influence the reported case count. The latter is especially true in the presence of underascertainment.

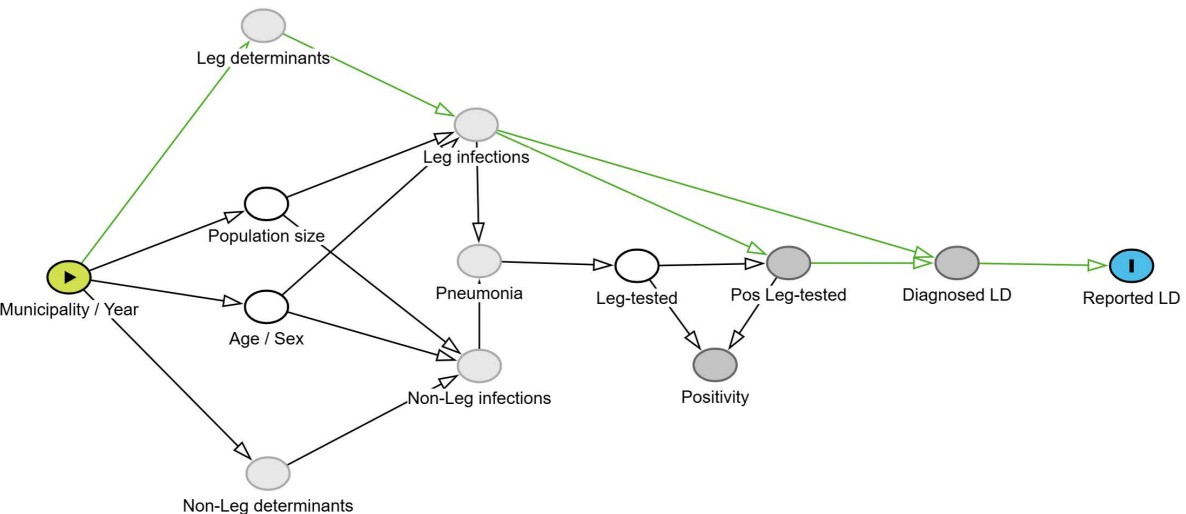

**Fig 1.  Directed Acyclic Graph (DAG) and direct effect on case reporting (estimand = green arrows, Leg = *Legionella* spp. or *L.pneumophila*, LD = Legionnaires' disease).**

The DAG further expresses our believe that, at population level, the volume of diagnostic Legionella testing is not merely a consequence of people having Legionnaires' disease, consulting a physician and receiving a diagnostic test. Instead, due to the aspecificity of Legionnaires' disease symptoms, we assume that the Legionella testing intensity is also (and to a larger extent) driven by the much larger volume of non-Legionella pneumonia, mainly community acquired pneumonia (CAP), and the clinicians' inclination to test CAP patients for Legionella. This assumption is supported by the higher volume of testing (Table 1, Table 2) and the more similar seasonality to non-LD CAP than to Legionnaires' disease (S2 Fig 4 in S1 Appendix) [48,49]. The effect of age and sex on Legionnaires' disease has been widely reported [23].

A difference between the count of persons tested positive for Legionella and persons diagnosed as Legionnaires' disease can be explained by elements of differential diagnosis other than clinical presentation of pneumonia and the Legionella test result. Such aspects can be symptomatology (e.g., presence or absence of high fever or bradycardia), clinical evolution (e.g., response to initial empirical therapy), blood chemistry values (e.g., presence or absence of hypophosphatemia or hyponatremia), lung radiography, *Legionella* test type (e.g., PCR *Legionella species* (*Legionella* spp.) versus *Legionella pneumophila* (*L.pn.*)), sample type (e.g., upper- vs. lower-respiratory tract sample), or likelihood of competing diagnoses (e.g., *Chlaymydia* or *Mycoplasma* infection, Q fever, psittacosis).

In principle, the count of diagnosed patients may differ from the count of reported cases due to imperfect compliance to report by physicians and labs. However, this is assumed negligible in Denmark since 2014 due to an automated detection algorithm on the Danish Microbiology Database, whereby for each positive test, the diagnosis is verified with the clinician.

**Table 1. Tested persons (n = 257 726) by age group, sex or province, and by year.**

| | 2014 (N=16970) | 2015 (N=21541) | 2016 (N=25221) | 2017 (N=26187) | 2018 (N=29359) | 2019 (N=28285) | 2020 (N=31327) | 2021 (N=37955) | 2022 (N=40881) |
|---|---|---|---|---|---|---|---|---|---|
| **Age group** | | | | | | | | | |
| 00-49 | 3121 (18.4%) | 4492 (20.9%) | 5732 (22.7%) | 5175 (19.8%) | 4968 (16.9%) | 5013 (17.7%) | 4948 (15.8%) | 6289 (16.6%) | 6468 (15.8%) |
| 50-59 | 2265 (13.3%) | 2781 (12.9%) | 3261 (12.9%) | 3237 (12.4%) | 3545 (12.1%) | 3456 (12.2%) | 3614 (11.5%) | 4114 (10.8%) | 4191 (10.3%) |
| 60-69 | 3982 (23.5%) | 4772 (22.2%) | 5251 (20.8%) | 5305 (20.3%) | 5974 (20.3%) | 5575 (19.7%) | 5780 (18.5%) | 6725 (17.7%) | 7186 (17.6%) |
| 70-79 | 4171 (24.6%) | 5393 (25.0%) | 6235 (24.7%) | 6862 (26.2%) | 8140 (27.7%) | 7771 (27.5%) | 8748 (27.9%) | 10347 (27.3%) | 11474 (28.1%) |
| over 80 y | 3431 (20.2%) | 4103 (19.0%) | 4742 (18.8%) | 5608 (21.4%) | 6732 (22.9%) | 6470 (22.9%) | 8237 (26.3%) | 10480 (27.6%) | 11562 (28.3%) |
| **Sex** | | | | | | | | | |
| female | 7894 (46.5%) | 10372 (48.2%) | 12255 (48.6%) | 12376 (47.3%) | 14011 (47.7%) | 13433 (47.5%) | 14409 (46.0%) | 17820 (47.0%) | 19697 (48.2%) |
| male | 9076 (53.5%) | 11169 (51.9%) | 12966 (51.4%) | 13811 (52.7%) | 15348 (52.3%) | 14852 (52.5%) | 16918 (54.0%) | 20135 (53.0%) | 21184 (51.8%) |
| **Province of residence** | | | | | | | | | |
| Bornholm | 78 (0.5%) | 117 (0.5%) | 226 (0.9%) | 165 (0.6%) | 132 (0.4%) | 151 (0.5%) | 174 (0.6%) | 226 (0.6%) | 291 (0.7%) |
| Copenhagen City | 2523 (14.9%) | 3531 (16.4%) | 4185 (16.6%) | 3977 (15.2%) | 3842 (13.1%) | 4166 (14.7%) | 4178 (13.3%) | 6333 (16.7%) | 6902 (16.9%) |
| Copenhagen Surroundings | 1345 (7.9%) | 1693 (7.9%) | 2128 (8.4%) | 2479 (9.5%) | 3090 (10.5%) | 3952 (14.0%) | 5462 (17.4%) | 6069 (16.0%) | 6073 (14.9%) |
| East Jutland | 1926 (11.3%) | 2680 (12.4%) | 3387 (13.4%) | 3675 (14.0%) | 3999 (13.6%) | 2853 (10.1%) | 2371 (7.6%) | 2598 (6.8%) | 2902 (7.1%) |
| East Zealand | 636 (3.7%) | 796 (3.7%) | 907 (3.6%) | 974 (3.7%) | 1187 (4.0%) | 830 (2.9%) | 654 (2.1%) | 1091 (2.9%) | 1533 (3.8%) |
| Funen | 2044 (12.0%) | 2773 (12.9%) | 3106 (12.3%) | 3463 (13.2%) | 3346 (11.4%) | 3870 (13.7%) | 5106 (16.3%) | 5209 (13.7%) | 4548 (11.1%) |
| North Jutland | 1094 (6.4%) | 1323 (6.1%) | 1488 (5.9%) | 1304 (5.0%) | 1368 (4.7%) | 1279 (4.5%) | 1024 (3.3%) | 1043 (2.7%) | 1236 (3.0%) |
| North Zealand | 2357 (13.9%) | 2159 (10.0%) | 1611 (6.4%) | 2155 (8.2%) | 2972 (10.1%) | 3185 (11.3%) | 4101 (13.1%) | 4852 (12.8%) | 4959 (12.1%) |
| South Jutland | 2326 (13.7%) | 2792 (13.0%) | 3411 (13.5%) | 3917 (15.0%) | 4618 (15.7%) | 4516 (16.0%) | 5269 (16.8%) | 5775 (15.2%) | 6926 (16.9%) |
| West and South Zealand | 1564 (9.2%) | 1826 (8.5%) | 2199 (8.7%) | 2392 (9.1%) | 2957 (10.1%) | 2091 (7.4%) | 1550 (4.9%) | 2505 (6.6%) | 3060 (7.5%) |
| West Jutland | 1077 (6.3%) | 1851 (8.6%) | 2573 (10.2%) | 1686 (6.4%) | 1848 (6.3%) | 1392 (4.9%) | 1438 (4.6%) | 2254 (5.9%) | 2451 (6.0%) |

Count (Column Percentage).

**Table 2. Reported cases (n = 2081) by age group, sex or province, and by year.**

| | 2014 (N = 147) | 2015 (N = 174) | 2016 (N = 161) | 2017 (N = 270) | 2018 (N = 258) | 2019 (N = 259) | 2020 (N = 268) | 2021 (N = 270) | 2022 (N = 274) |
|---|---|---|---|---|---|---|---|---|---|
| **Age group** | | | | | | | | | |
| 00-49 | 15 (10.2%) | 11 (6.3%) | 22 (13.7%) | 22 (8.1%) | 23 (8.9%) | 24 (9.3%) | 25 (9.3%) | 23 (8.5%) | 18 (6.6%) |
| 50-59 | 29 (19.7%) | 25 (14.4%) | 30 (18.6%) | 53 (19.6%) | 46 (17.8%) | 37 (14.3%) | 41 (15.3%) | 38 (14.1%) | 24 (8.8%) |
| 60-69 | 49 (33.3%) | 53 (30.5%) | 48 (29.8%) | 72 (26.7%) | 62 (24.0%) | 66 (25.5%) | 57 (21.3%) | 58 (21.5%) | 59 (21.5%) |
| 70-79 | 32 (21.8%) | 53 (30.5%) | 40 (24.8%) | 66 (24.4%) | 75 (29.1%) | 73 (28.2%) | 79 (29.5%) | 79 (29.3%) | 98 (35.8%) |
| over 80 y | 22 (15.0%) | 32 (18.4%) | 21 (13.0%) | 57 (21.1%) | 52 (20.2%) | 59 (22.8%) | 66 (24.6%) | 72 (26.7%) | 75 (27.4%) |
| **Sex** | | | | | | | | | |
| female | 57 (38.8%) | 60 (34.5%) | 60 (37.3%) | 103 (38.1%) | 90 (34.9%) | 103 (39.8%) | 114 (42.5%) | 105 (38.9%) | 114 (41.6%) |
| male | 90 (61.2%) | 114 (65.5%) | 101 (62.7%) | 167 (61.9%) | 168 (65.1%) | 156 (60.2%) | 154 (57.5%) | 165 (61.1%) | 160 (58.4%) |
| **Province of residence** | | | | | | | | | |
| Copenhagen City and Bornholm | 16 (10.9%) | 11 (6.3%) | 17 (10.6%) | 12 (4.4%) | 17 (6.6%) | 16 (6.2%) | 9 (3.4%) | 35 (13.0%) | 36 (13.1%) |
| Copenhagen Surroundings | 12 (8.2%) | 13 (7.5%) | 9 (5.6%) | 19 (7.0%) | 30 (11.6%) | 33 (12.7%) | 39 (14.6%) | 41 (15.2%) | 35 (12.8%) |
| East Jutland | 18 (12.2%) | 29 (16.7%) | 27 (16.8%) | 62 (23.0%) | 38 (14.7%) | 29 (11.2%) | 23 (8.6%) | 15 (5.6%) | 17 (6.2%) |
| East Zealand | 7 (4.8%) | 12 (6.9%) | 5 (3.1%) | 12 (4.4%) | 17 (6.6%) | 12 (4.6%) | 13 (4.9%) | 13 (4.8%) | 17 (6.2%) |
| Funen | 26 (17.7%) | 40 (23.0%) | 28 (17.4%) | 40 (14.8%) | 30 (11.6%) | 49 (18.9%) | 45 (16.8%) | 30 (11.1%) | 34 (12.4%) |
| North Jutland | 13 (8.8%) | 8 (4.6%) | 17 (10.6%) | 16 (5.9%) | 18 (7.0%) | 16 (6.2%) | 18 (6.7%) | 18 (6.7%) | 13 (4.7%) |
| North Zealand | 17 (11.6%) | 16 (9.2%) | 7 (4.3%) | 29 (10.7%) | 26 (10.1%) | 31 (12.0%) | 35 (13.1%) | 31 (11.5%) | 45 (16.4%) |
| South Jutland | 15 (10.2%) | 21 (12.1%) | 27 (16.8%) | 39 (14.4%) | 47 (18.2%) | 37 (14.3%) | 44 (16.4%) | 41 (15.2%) | 34 (12.4%) |
| West and South Zealand | 13 (8.8%) | 17 (9.8%) | 13 (8.1%) | 27 (10.0%) | 22 (8.5%) | 21 (8.1%) | 25 (9.3%) | 36 (13.3%) | 31 (11.3%) |
| West Jutland | 10 (6.8%) | 7 (4.0%) | 11 (6.8%) | 14 (5.2%) | 13 (5.0%) | 15 (5.8%) | 17 (6.3%) | 10 (3.7%) | 12 (4.4%) |

Count (Column Percentage).

## b) Data

The data concerned all Legionella tests performed in Denmark between 2014 and 2022, retrieved from the Danish Micro-biology Database (MiBa) [50–52]. Age, sex, and municipality of residence from the tested persons originated from the civil registration. Further, the study included data of all Legionnaires' disease cases reported between 2014 and 2022 to the Department of Infection Epidemiology and Prevention at Statens Serum Institut [53].

Samples and data had been collected as part of routine national Legionnaires' disease surveillance and were exempt from informed consent procedures. The data (2014–2021) were made available for research purpose as of May 2022, and updated data (to include 2022) as of May 2023. Data were pseudonymized and aggregated for the analysis. No individuals could be identified with the data available to the analyst. According to Danish law, no approval from an ethics committee was required for this surveillance study. This study was performed under the auspices of Statens Serum Institut as per the Danish Health Act, Section 222 [54]. Permission to use the data for this report was granted by Data Protection and Information Security (project J. No. 21/06213, Legionnaires' disease diagnostics; testing intensity and case reporting for Legionnaires' disease in Denmark 2014–2022).

The tests included were urinary antigen test for *L. pneumophila* SG1 (UAT), polymerase chain reaction test for *L. pneumophila* (PCR *L. pneumophila*) or PCR for *Legionella* spp. (PCR *Legionella* spp.). Tests were performed on urine samples or -upper or lower- respiratory tract samples. We used the testing date, the municipality of residence at the time of the test, sex, and five-year age group at time of test, estimated from mid-year of birth.

Annual tested persons were defined as persons who had received at least one of the three types of Legionella tests in a given year. Some individuals were tested in several years; presumably for different pneumonia episodes. However, a small proportion who were tested at the end and start of a year may have been double-counted while referring to the same disease episode. We defined the within-year test profile as the combination of test types a person received in a given year, and report these for descriptive purpose only.

Cases fulfilled the EU case definition of confirmed or probable case. Throughout the study period, the EU case definition considered a positive urinary antigen test as a confirmed case criterium, and a positive PCR test as a probable case criterium, but this distinction in case classification was not deemed relevant to our study and was therefore not used. The setting of exposure was travel-, healthcare-, or community-associated, all contributing to the total case counts, i.e., without differentiating whether a case may have been infected locally or abroad, in the community or in a hospital, or was part of an outbreak or not. From the cases, we used the reporting date, as well as the sex and five-year age group at time of report. Case location was, as for tested persons, the municipality of residence at time of the (positive) test, retrieved from the microbiology database and the civil registration. Cases for which the location could not be retrieved (102 cases, evenly distributed over the nine study years), were included in the description but excluded for the spatial modeling.

Demographic data about the population by municipality (N = 99), year (9), sex (2), and 5-year age groups (19) were downloaded from Statistics Denmark [55]. We applied indirect standardization based on the entire country and study period as reference to obtain expected counts.

We defined two derived quantities. Diagnostic testing intensity (rate) was defined as the number of annual tested persons per 100,000 inhabitants per year. Case reporting (rate) was defined as the number of cases reported to the Epidemiological Surveillance System per 100,000 inhabitants per year.

## c) Statistical model fitting and model comparison

To obtain estimates of diagnostic testing intensity and of case reporting by geographical area and time, we built Bayesian hierarchical spatio-temporal models. We modelled yearly data at municipality level. We used models that allow for partial pooling of municipal data [47,56] as small areas have due to their limited population size typically highly variable standardized incidence ratios (SIR), making these inappropriate for our purpose.

Considering our DAG and estimand, we selected the model's functional form (not expressed by the DAG) based on the lowest widely applicable information criterion (WAIC) [47,57]. We explored models with a variety of structured and unstructured spatial effects, fixed or varying time effects, and space-time interaction effects. Non-linear effects were modelled as random walk (RW) and autoregressive (AR) processes. Several spatial and spatio-temporal models were compared. All fitted models are listed in supplement (S9, S10 in S1 Appendix).

Models were fitted using Integrated Nested Laplace Approximation, INLA, an approximate Bayesian inference framework for latent Gaussian models [58]. Analyses were performed using R Statistical Software, version 4.4.0 [59] and the R-INLA package [60]. Graphs and maps of the results were created with the R packages ggplot2 version 3.5.1 and tmap version 4.2. The basemap of the Danish municipalities was adapted from OpenStreetMap, available under the Open Database License (https://www.openstreetmap.org/copyright). Computing code for modeling and visualisations from paper and supplement is available at GitHub (https://github.com/erobesyn/dk_testing_reporting) and a permanently archived Zenodo repository (https://doi.org/10.5281/zenodo.15238532).

The best fitting models, both for the diagnostic testing intensity and for the case reporting, have a spatial convolution effect composed of a spatially correlated effect (ICAR) and spatially uncorrelated effect (iid) (as in Besag, York and Molie model) [61,62], a temporal random walk of first order, and an unstructured spatio-temporal interaction (iid). This model is known as type I Knorr-Held model [63].

The model for the estimated count of tested persons (exposure model) is given in equation (A):

$$T_{it} \sim \text{Poisson}(\mu_{it})$$

$$\mu_{it} = e_{it}\, r_{it}$$

$$\log(r_{it}) = \alpha_0 + \xi_i + \gamma_t + \delta_{it}$$

where i represents the municipality and t the year.

The model for the estimated count of reported cases (outcome model) is defined in equation (B):

$$C_{it} \sim \text{Poisson}(\mu_{it})$$

$$\mu_{it} = e_{it}\,r_{it}$$

$$\log(r_{it}) = \alpha_0 + \beta_{T.obs_{it}} + \xi_i + \gamma_t + \delta_{it}$$

where i represents the municipality and t the year.

Further, $e_{it}$ is the 5-year age group- and sex-standardized expected count, and $r_{it}$ is the relative risk of interest. In the Results section, we call the relative risk from the two models, respectively, the 'relative testing intensity ($RR_T$)' and the 'relative case reporting ($RR_C$)', i.e., the multiplicative factor for a specific year and municipality as compared to the expected value. A value of two means that the testing intensity or case reporting was twice as large as expected from the overall intensity/reporting in the whole country.

In the model, $\alpha_0$ is an intercept, $\beta_{T.obs_{it}}$ a varying effect of binned count of tested persons, $\xi_i$ a spatial varying effect, $\gamma_t$ a temporal varying effect, and $\delta_{it}$ a spatio-temporal interaction effect. The spatial varying effect $\xi_i$ is composed of a spatially structured and a spatially unstructured effect (S8 in S1 Appendix).

## Results

### a) Description

In the study period between 2014 and 2022, the Danish population has grown monotonically from 5.63 to 5.87 million inhabitants. Our analysis did not include Greenland and the Faroe Islands, with population sizes in 2022 of 56,661 and 53,952, respectively.

The proportion of Danish people with age over 70 years has increased in our study period from 11.9% to 14.9% (+3 percentage points), while the national population's sex ratio remained roughly the same with around 49.7% male (S5 Table 2 in S1 Appendix). Bornholm is the province with the smallest population, i.e., 39,638 inhabitants (2022), while East Jutland is the most populated province with 913,861 inhabitants (2022). The median population in 2022 by municipality was 43,089 (IQR 30,332–59,752), ranging from 93 (Christiansoe) to 644,431 (Copenhagen).

Excluding 0.3% of performed tests for which the result was not registered, we included 350 814 tests with positive or negative result and for which age, sex, and municipality of residence of the tested person was known. Of these, 4,752 (1.35%) were positive tests. The yearly number of tests performed in Denmark more than doubled over the study period, from 24,065 tests (2014) to 54,928 tests (2022). The relative proportions of urinary antigen test and PCR changed over time: the proportion UAT decreased from 39% in 2014 to 11% in 2022, while the proportion PCR *L. pneumophila* increased from 44% to 71%. The proportion PCR *Legionella* spp. remained stable at around 17%.

257,729 persons were tested for *Legionella* infection in a given year (annual tested persons), corresponding to 214,807 unique tested persons over the nine-year study period. Of the annual tested persons, 2,634 (1.0%) were found to have tested positive (in one or more of their tests).

The number of annual tested persons grew steadily from 16,970 tested persons (2014) to 40,881 tested persons (2022) (Table 1, and S7 Fig 9 in S1 Appendix). The proportion of tested persons with age over 70 years has increased from 44.8% to 56.4% (+11.6 percentage points). The proportion of tested males fluctuated between 51.4% and 54.0% without

clear year trend. The annual number of tested persons per province ranged, in 2022, from 291 (Bornholm) to 6926 (South Jutland) (S7 Fig 10 in S1 Appendix). The median number of tested persons in 2022 by municipality was 305 (IQR 158–498), ranging from less than ten (Christiansoe and Laesoe) to 5142 (Copenhagen).

Furthermore, the annual tested persons can be grouped by their within-year test profile (S6 Table 3 in S1 Appendix). The proportion of tested persons who received, within a given year, only UAT without PCR decreased from 27% in 2014 to 6.2% in 2022. Also, the test profile UAT with PCR *Lpn* decreased from 17.8% in 2014 to 5.2% in 2022. On the other hand, the proportion of tested persons who received within a given year only PCR *Lpn* doubled from 33.9% in 2014 to 67.8% in 2022, with the largest increase (+17 percentage points) in 2020, the first year of the COVID-19 pandemic.

In the nine year period, there were 2,183 reported Legionnaires' disease cases, of which 2,081 with the location data needed for spatial description and modeling. The 2,183 cases correspond to 83% of all positive tested persons in the database. They consist of 63.2% community acquired cases (CALD, n = 1,380), 10.0% healthcare associated cases (HALD, n = 218), 18.8% travel-associated cases (TALD, n = 410), and 8.0% cases for which the setting was not recorded (n = 175).

The number of reported cases with known location has overall grown from 147 cases in 2014 to 274 in 2022, with the largest increase from 2016 to 2017, when 109 extra cases (+68%) were reported (Table 2, and S7 Fig 9 in S1 Appendix). The proportion of reported cases with age over 70 years grew from 36.8% to 63.2% (+26.4 percentage points), however with yearly fluctuations. The proportion male fluctuated between 57.5 and 65.5% without clear trend. The annual number of reported cases per province ranged, in 2022, from well below ten (Bornholm) to 45 (North Zealand) but reached 62 in 2017 in East Jutland (S7 Fig 11 in S1 Appendix). The median number of reported cases in 2022 by municipality was 2 (IQR 1–4); for all municipalities there were less than 10 cases in 2022, except for Frederiksberg, Odense and Copenhagen with respectively 10, 13, and 22 cases.

## b) Diagnostic testing intensity

The estimated diagnostic Legionella testing intensity is determined by the age- and sex-standardized relative testing intensity estimated from our model and the population size, and age and sex distribution in a given municipality. Fig 2 shows the map of the testing intensity by year and municipality. The municipal testing intensity ranged from 128 to 2,446 per 100,000 inhabitants. The median municipal testing intensity over the entire nine-years study period was 419 tested persons per 100,000 inhabitants per year (IQR 294–631). The median increased steadily from 275 tested persons per 100,000 inhabitants in 2014 to 620 in both 2021 and 2022. Of the ten lowest testing intensities, nine are from municipalities in North Jutland, mainly in 2014. Since 2020, some municipalities reached testing intensities of over 1,500 per 100,000 inhabitants, i.e., testing 1 in 67 inhabitants in a given year. This was the case in nine municipalities: five in Funen (Kerteminde, Langeland, Nyborg, Svendborg, Faaborg-Midtfyn), three in South Jutland (Vejen, Billund, Esbjerg), and one in Copenhagen Surroundings (Broendby). Among these, the most intensive testing took place in Langeland and Vejen, where more than 1 in 50 inhabitants received at least one of the three *Legionella* tests in a given year. The testing intensity in Langeland increased gradually to be in 2022 six-fold the level of 2014.

The estimated age- and sex-standardized relative testing intensity in municipalities ($RR_T$), ranged from 0.18 (Laesoe, North Jutland, 2014) to 4.01 (Vejen, South Jutland, 2022) (S11 Fig 12–13 in S1 Appendix). The highest relative testing intensities, with $RR_T$ over 3, were all in 2021 and 2022, either in South Jutland (Vejen, Esbjerg and Billund), Copenhagen Surroundings (Hvidovre) or Funen (Langeland, Nyborg). Twenty-six municipalities had a relative testing intensity over 2. The lowest relative testing intensities, with $RR_T$ under 0.33, were in North Jutland (Broenderslev, Frederikshavn, Hjoerring, Laesoe, Morsoe, Rebild, Thisted, Jammerbugt, Mariagerfjord) and South Jutland (Fanoe, 2014 only). Fifty-five municipalities had a relative testing intensity under 0.5.

The variance in relative testing intensities among municipalities can be attributed to different components. The largest component, 55.6% of the variance, is spatially structured variation related to the neighbourhood structure of municipalities,

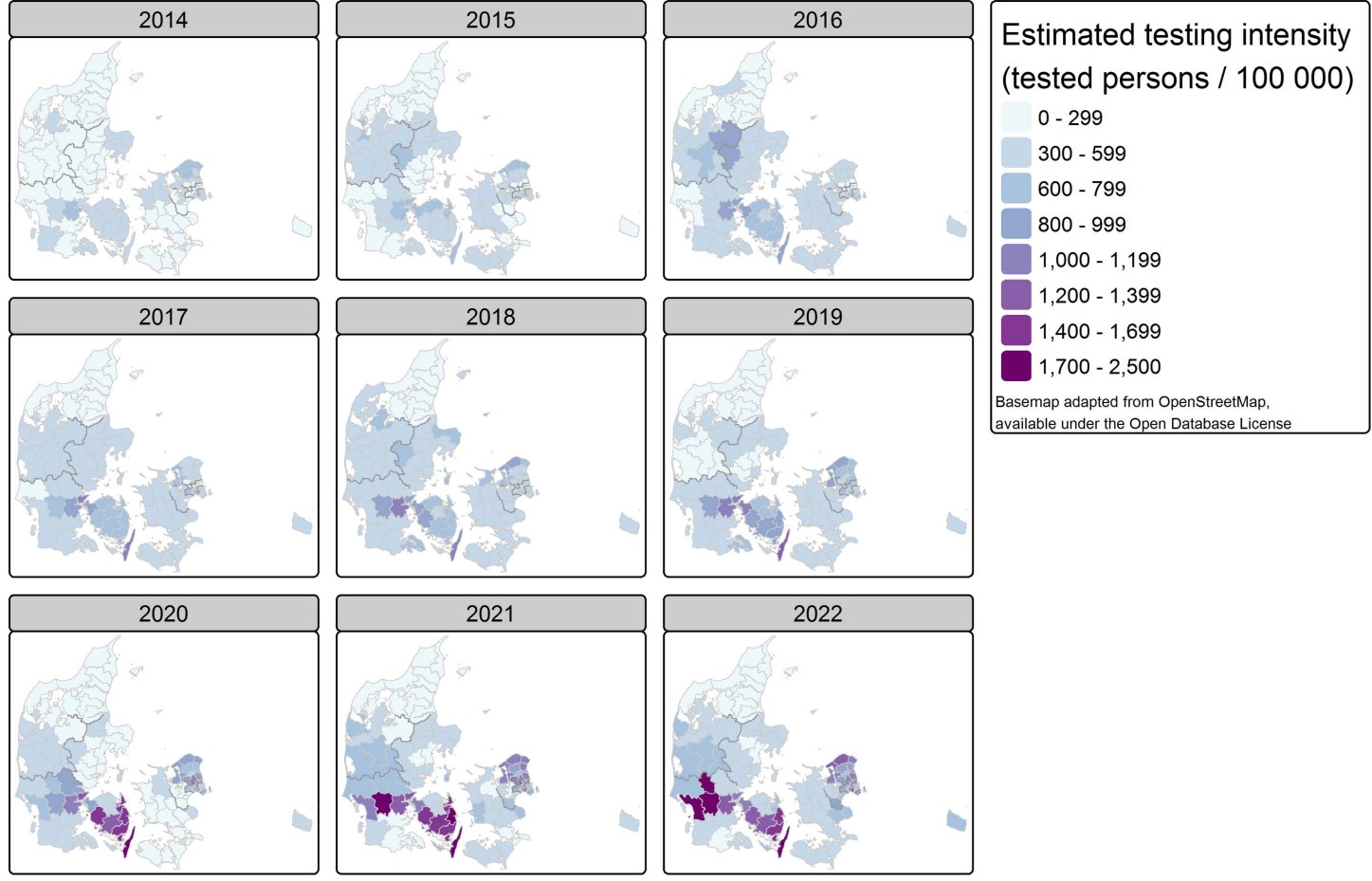

**Fig 2. Map of estimated testing intensity, by year and municipality, Denmark 2014-2022.**

expressing that bordering municipalities are more alike than far away municipalities. Further, 1.0% can be explained by spatially unstructured variation, 5.5% by temporal variation, and 37.9% by spatio-temporal interaction. The structured and unstructured spatial varying effects and spatio-temporal interaction varying effect are shown in the supplementary material (S12 Fig 15, S13 Fig 16 in S1 Appendix).

We distinguish an area with high exceedance probability, $Pr(RR_T > 1)$, i.e., where the testing intensity is very likely to be higher than expected (S11 Fig 14 in S1 Appendix). This area covers the majority of Funen, a triangular part of South and West Jutland, and the area of Copenhagen City and Surroundings. Complement to this, we identify Zealand, the south of South Jutland, East Jutland and North Jutland, as areas with a likely deficit in relative testing intensity.

#### c) Effect of diagnostic testing intensity on case reporting

We estimated that the relative Legionnaires' disease testing intensity in Denmark has steadily increased over time (Fig 3b). Further, we estimated that with increasing testing intensity, the relative case reporting increased (Fig 3a and S14 Fig 17 in S1 Appendix). The increasing relative case reporting is observed up to about 1000–1200 tested persons per 100 000 inhabitants, with relative case reporting of about 1.7. At higher testing intensities, there is no further improvement in

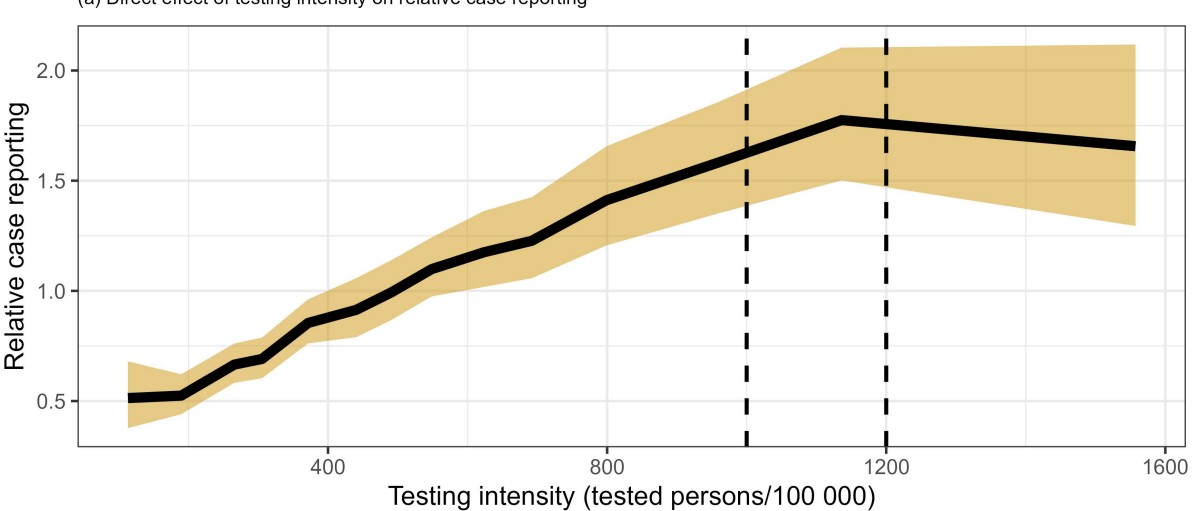

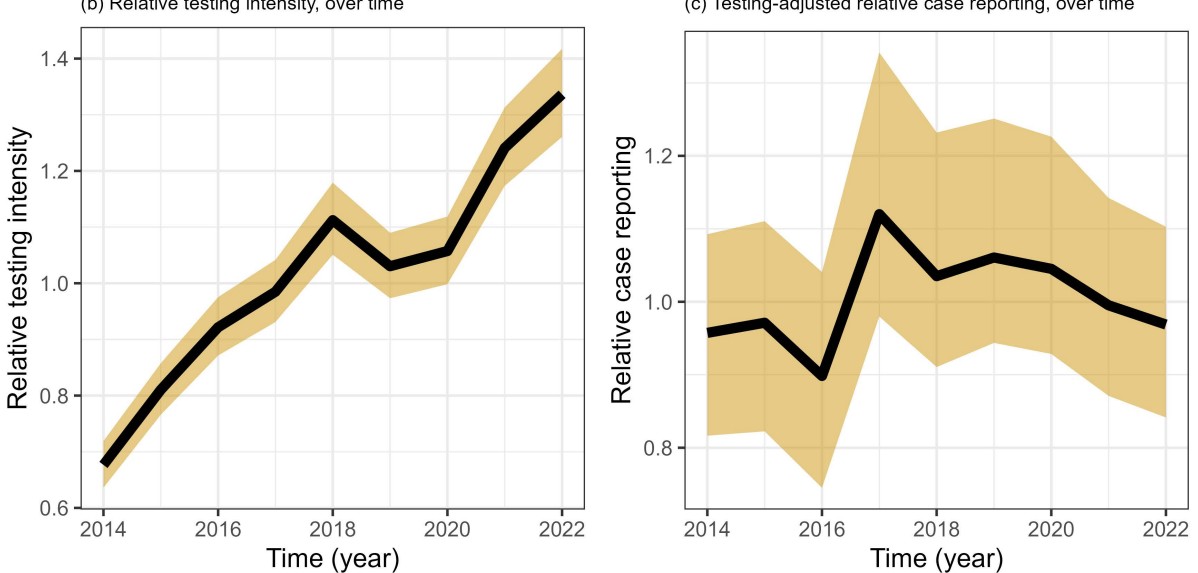

**Fig 3. Effect of diagnostic testing intensity on case reporting, and evolution over time of both.** (a) Effect of testing intensity on relative case reporting, (b) Evolution of relative testing intensity over time, (c) Evolution of relative case reporting, adjusted for testing intensity, over time. Effects (and 95% compatibility interval) on the relative risk scale. Denmark 2014–2022.

case reporting and there is larger uncertainty of the estimate. The relative Legionnaires' disease case reporting, when adjusted for the testing intensity, does not increase over time but shows a small dip in 2016, followed by a small peak in 2017 (Fig 3, c).

### d)  Case reporting, adjusted for varying testing intensity

We estimated the municipal, annual, testing-adjusted case reporting. The testing-adjusted case reporting ranged from 1.4 (Aarhus, East Jutland, 2021) to 12.0 (Langeland, Funen, 2020) per 100 000 inhabitants. The median over the entire study

period and all municipalities was 3.9 reported cases per 100 000 inhabitants (IQR 2.9–5.6). The annual median fluctuated over the study period between 2.5 (range 1.4–6.2) in 2014 and 5.2 (range 1.5–11.7) in 2022 with a flat overall time effect.

Fig 4 shows the evolution of the case reporting both over time and space. In two municipalities the incidence exceeded 11/100 000 in one or more years, namely Langeland (Funen) and Halsnaes (North Zealand). Twelve other municipalities had in one or more years a reporting incidence over 9/100 000, mainly in Funen (Middelfart, Nyborg, Svendborg, Assens, Faaborg-Midtfyn) and North Zealand (Hoersholm, Gribskov, Furesoe, Helsingoer), but also two municipalities in Copenhagen Surroundings (Herlev, Roedoevre) and one in South Jutland (Kolding). Langeland was the municipality with the longest sustained very high Legionnaires' disease incidence, with seven out of nine years exceeding an incidence of 9/100 000, followed by Nyborg which exceeded this threshold during five years.

In contrast, Aarhus and Skanderborg (East Jutland) both had less than 1.5 cases per 100 000 inhabitants: for Skanderborg only in 2014, but for Aarhus in 2014 as well as in 2021 and 2022. In 19 other municipalities the adjusted reporting was below 2 per 100 000 inhabitants in at least one year: in the province Bornholm (Bornholm), Copenhagen City (Copenhagen), Copenhagen Surroundings (Albertslund, Gentofte, Gladsaxe), North Jutland (Rebild, Thisted, Vesthimmerlands, Broenderslev), North Zealand (Egedal, Furesoe), East Jutland (Horsens, Odder), East-Zealand (Solroed), South Jutland (Billund, Esbjerg, Varde), and West Jutland (Ikast-Brande, Viborg).

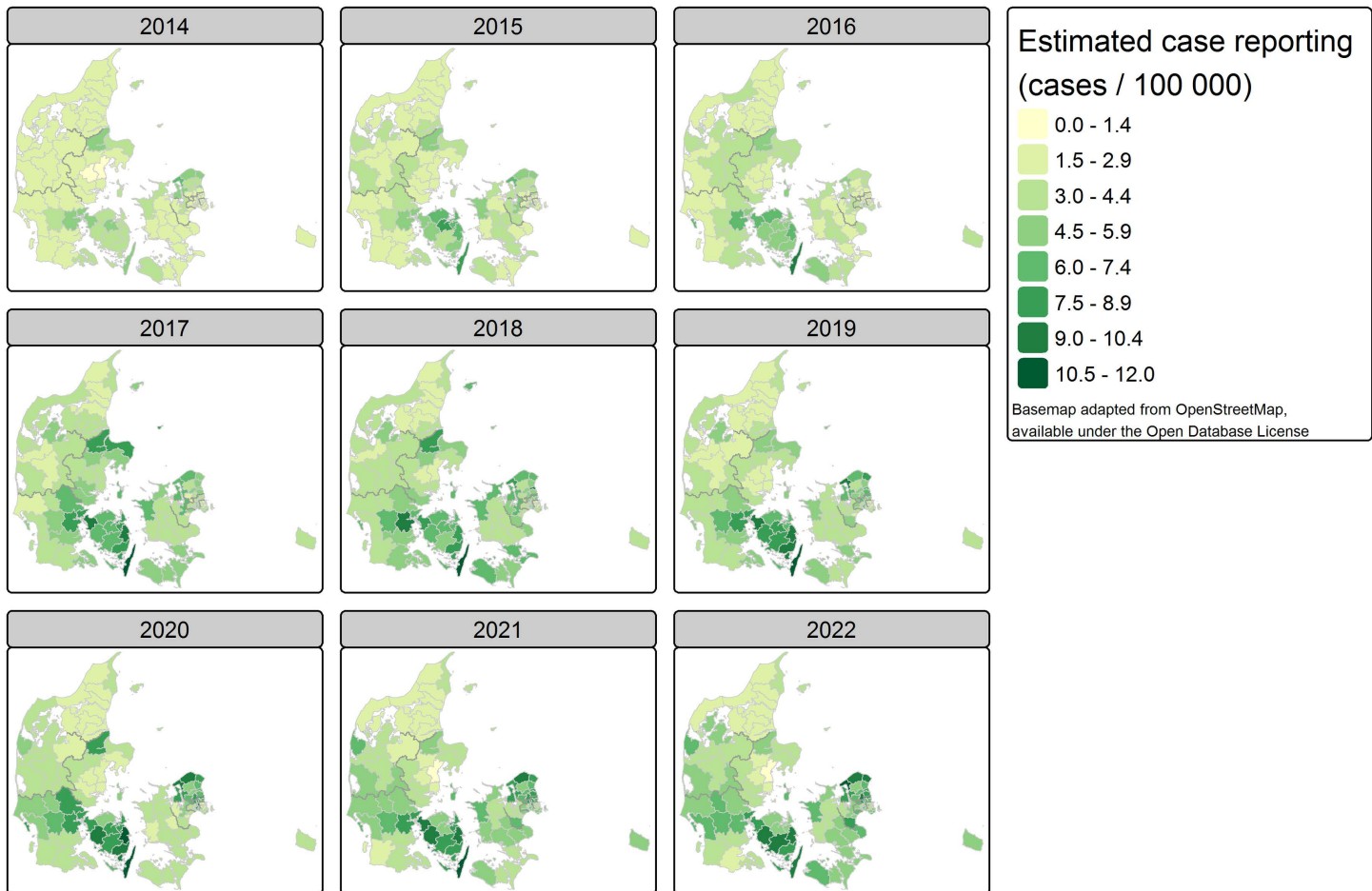

**Fig 4. Map of estimated case reporting, adjusted for varying testing intensity, by year and municipality, Denmark 2014-2022.**

The estimated age- and sex-standardized relative case reporting (RR$_C$), compared to the expected for the nine-year period, adjusted for testing intensity, ranged from 0.38 (Bornholm, Bornholm, 2014) to 2.76 (Roedovre, Copenhagen Surroundings, 2020) (Fig 5). Also, Herlev (Copenhagen Surroundings) had in 2020 more than 2.5 times the expected relative case reporting incidence. The mean relative case reporting exceeded the value of 2 in 2 other municipalities of Copenhagen Surroundings (Gladsaxe, Lyngby-Taarbaek), 5 municipalities of Funen (Assens, Middelfart, Nyborg, Odense, Svendborg), 2 municipalities of North Zealand (Furesoe, Halsnaes), 1 municipality of East Jutland (Randers), and 2 municipalities of South Jutland (Kolding, Vejle). Kolding and Odense were affected for the longest time, in respectively, 5 and 4 of the 9 years.

Twenty-six municipalities had (in least one year) a case reporting less than half the expected value (RR$_C$ < 0.5). Of these, the following municipalities had the lowest relative testing-adjusted case reporting (RR$_C$ ≈ 0.4): Bornholm (Bornholm), Varde (South Jutland), Skanderborg, Aarhus, Odder (East Jutland), and Vesthimmerlands and Broenderslev (North Jutland).

For the detection of LD clusters, we consider the probability that the relative case reporting is larger than one (Pr(RR$_C$)>1)(Fig 6), i.e., how likely is it that there are more than expected cases, accounting for varying age and sex distribution and varying testing levels across municipalities.

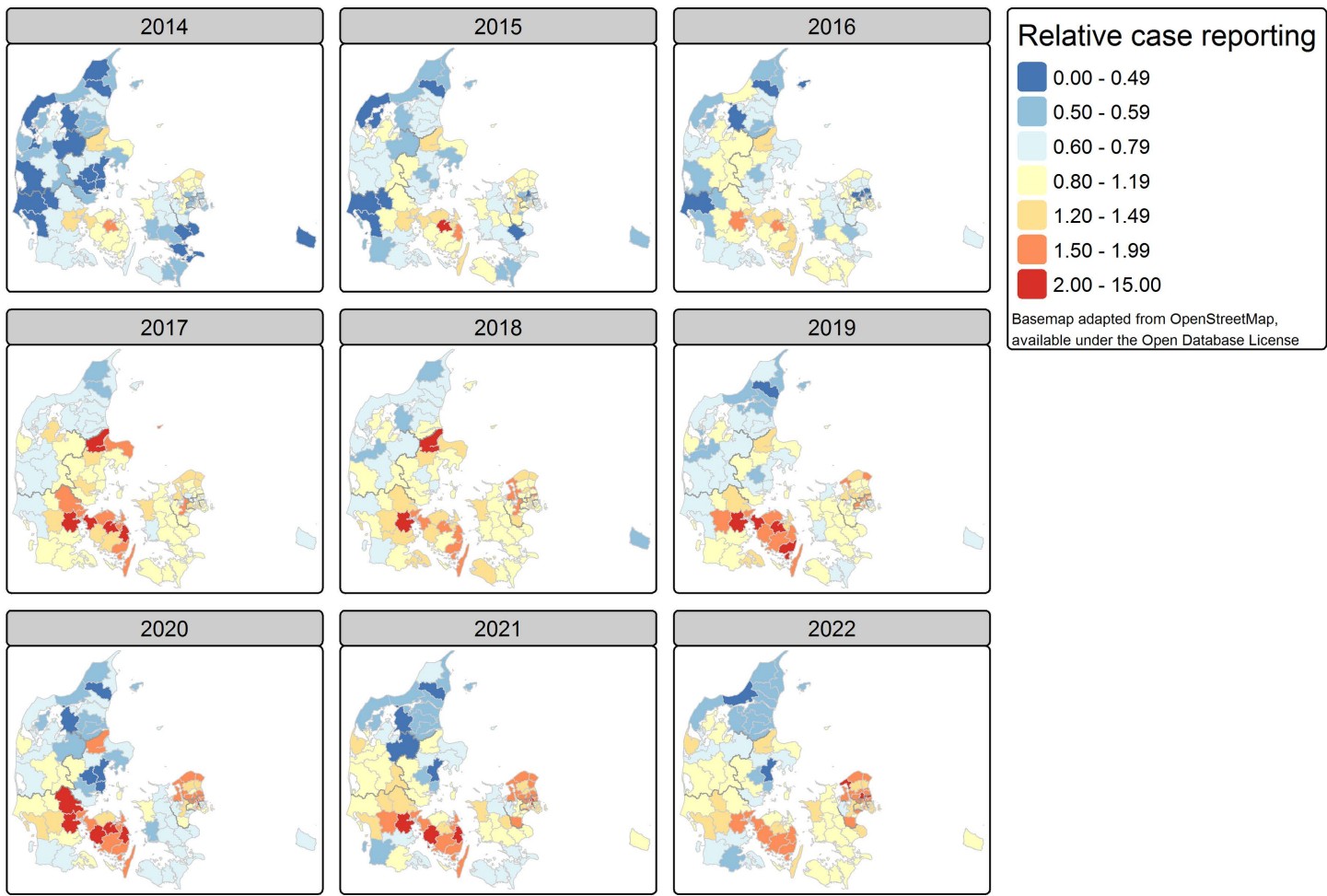

**Fig 5. Map of relative case reporting, adjusted for varying testing intensity, by year and municipality, Denmark 2014-2022.**

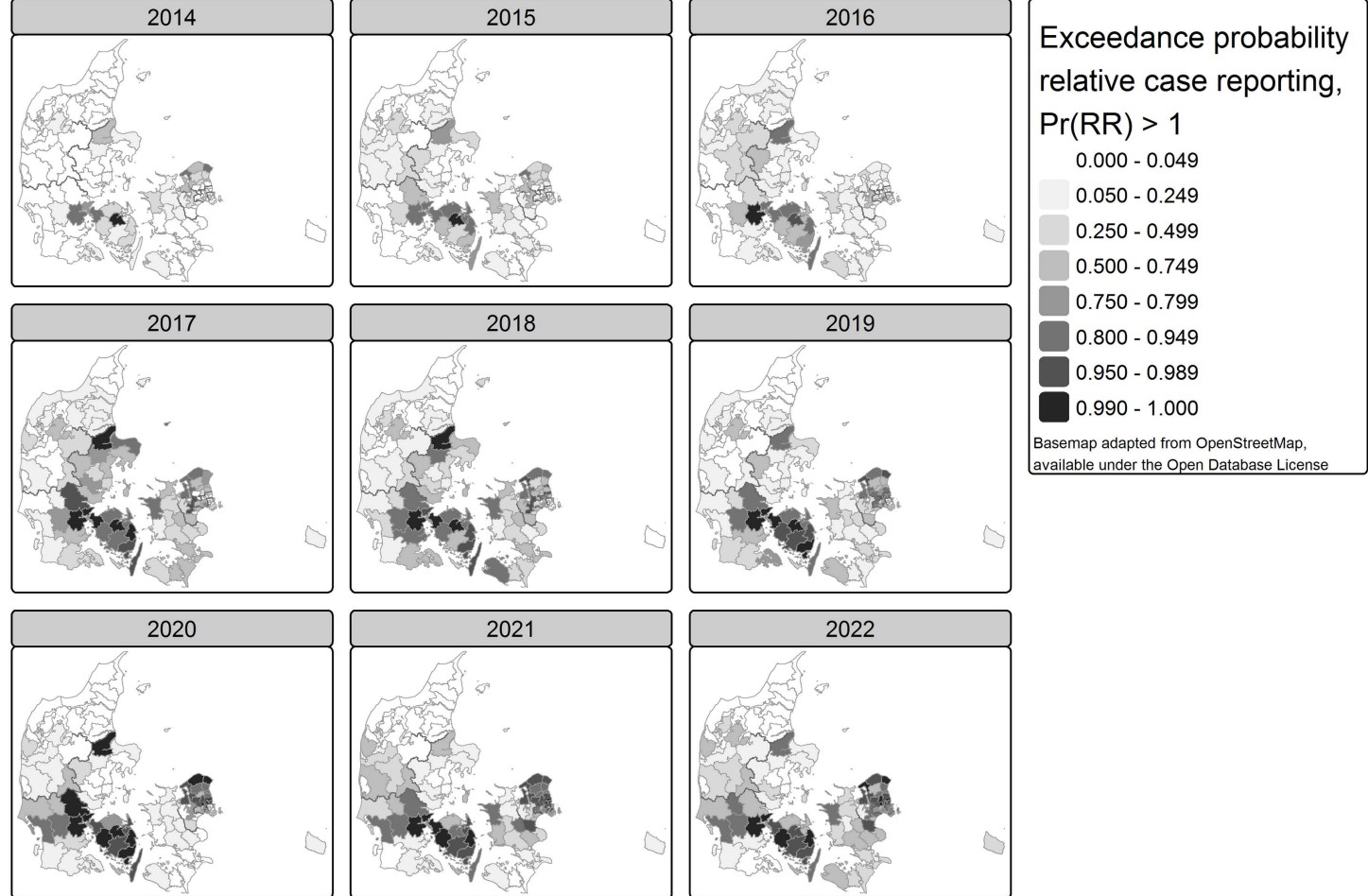

**Fig 6. Map of exceedance probability (Pr(RR > 1)) of relative case reporting, adjusted for varying testing intensity, by year and municipality, Denmark 2014-2022.**

Thirty-two municipalities had in at least one of the years, with over 95% probability, excess case reporting, compared to the nine-year expected value, and for 18 of these, this was nearly certain (over 99% probability): Odense, Middelfart, Nyborg, Svendborg, Assens, Ballerup, Gladsaxe, Herlev, Lyngby-Taarbaek, Roedovre, Gribskov, Helsingoer, Furesoe, Halsnaes, Randers, Kolding, Fredericia, and Vejle. Sixty-four municipalities had in at least one of the years over 95% probability to have deficit case reporting, and for 34, this was nearly certain (over 99% probability).

The variance in case reporting across the municipalities and years can be attributed to different components: 10.0% can be explained by spatially structured variation, 40.5% by spatially unstructured variation, 7.9% by temporal variation, and 41.6% by spatio-temporal interaction. The structured spatial effect shows strong homogeneity within regions, explained by some spatial similarity among municipalities, which can be environmental (e.g., weather, soil, housing water systems) or related to distribution of certain Legionella sequence types (and their pathogenicity). However, the spatial structured effect contributes proportionally less to the overall variation than we saw for testing intensity. In supplement, the reader will find the maps for these varying effects (S16 Fig 19, S17 Fig 20 in S1 Appendix).

Apart from the spatio-temporal excess relative case reporting, the space-time interaction can be considered a complementary indication of disease clustering in a given area and year [56,64]. Fig 7 shows the probability that such space-time

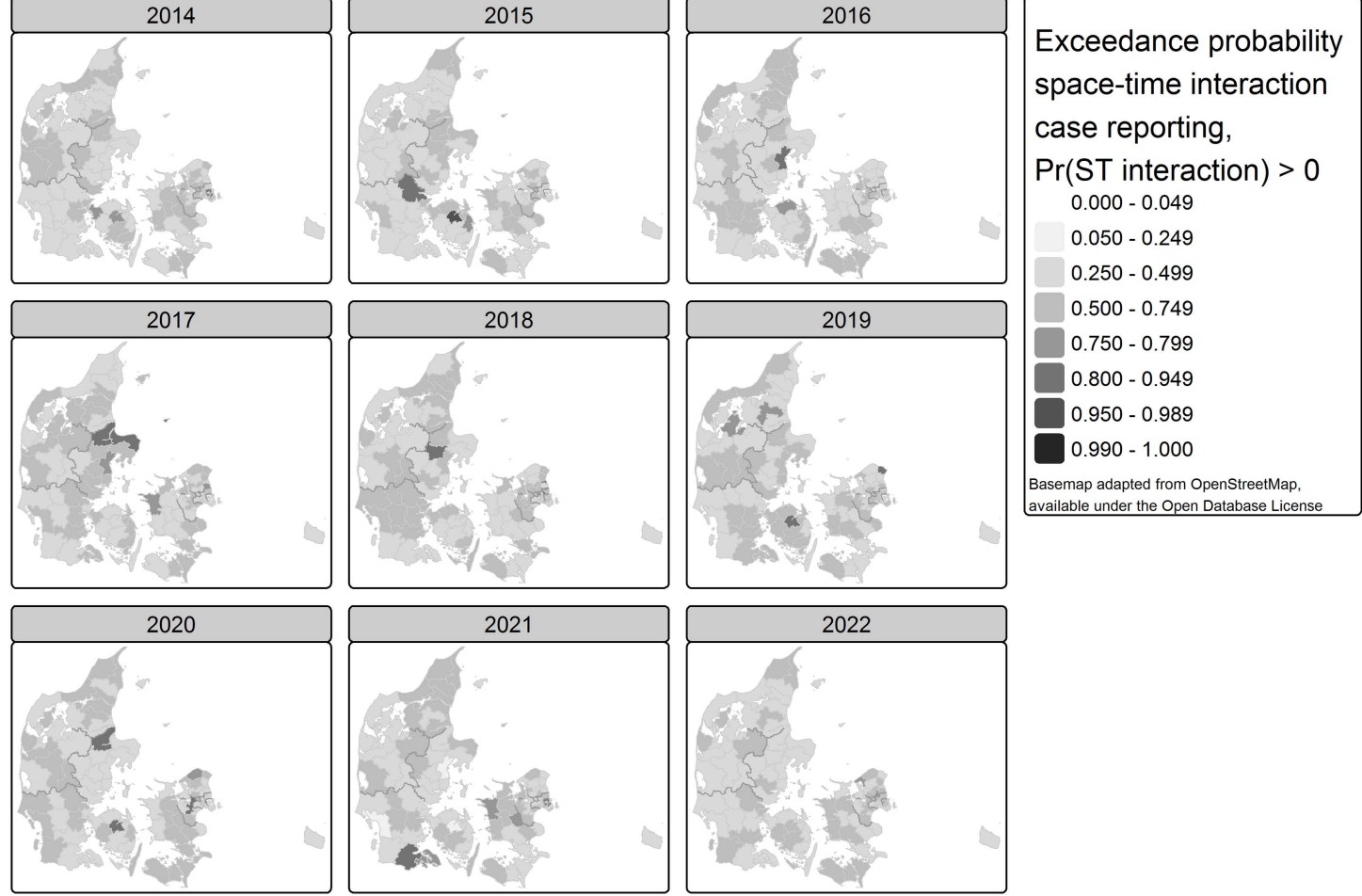

**Fig 7. Map of exceedance probability (Pr(STinteraction>0)) of spatio-temporal interaction of case reporting, adjusted for varying testing intensity, by year and municipality, Denmark 2014-2022.**

interaction is larger than zero. These are areas and timepoints with high probability that some clustering process has happened beyond the age, sex, and testing adjusted spatio-temporal trend. We found that only in Odense (Fyn) in 2015 the space-time interaction was beyond the expected with a probability over 95%.

## Discussion

In our study, we quantified the intensity of Legionnaires' disease diagnostic testing in all Danish municipalities between 2014 and 2022 and showed that within Denmark there was substantial variation, both spatially and over time, with an overall increase from 275 tested persons per 100,000 inhabitants in 2014 to 620 tested persons per 100 000 inhabitants in 2022. Our estimation is based on comprehensive national test data of *Legionella* urinary antigen test, PCR for *Legionella* spp. and PCR for *L. pneumophila* for residents of Denmark.

Whereas the population of Denmark between 2014 and 2022 has increased with 4%, the number of persons tested for Legionella in a given year has increased with 141%. While diagnostic testing intensity may not be known, a similar pattern may take place in other countries. Many reports and studies signaling apparent increased Legionnaires' disease incidence without considering the testing intensity will therefore not reliably reflect the underlying Legionnaires' disease incidence.

Whereas since the beginning of the century urinary antigen test availability drastically has changed the diagnostic landscape, we described during our study period a shift from urinary antigen test use to PCR use, further boosted by PCR lab capacity created during the COVID-19 pandemic. We considered in our study the total testing intensity and did not focus on the test characteristics of the urinary antigen and PCR tests, but these are also key to understand the yield of the tests and thus the resulting reporting incidence.

Over half of the variance in relative testing intensity is found to be spatially structured, with strong homogeneity within provincial boundaries. We expect this spatial structure to be attributable to the influence of clinical laboratories on diagnostic testing although assessing this was outside the scope of this study and would require sufficiently detailed information about the dynamic landscape of clinical laboratories' action radii.

We also quantified the positive causal effect of diagnostic testing intensity on the case reporting incidence under the mandatory notification surveillance system. We found evidence for a threshold of yearly diagnostic testing intensity of slightly over 1% of the overall population, above which the disease detection does not increase anymore. This reflects a potential optimal level of diagnostic testing from a disease surveillance perspective, but is also relevant for clinical practice. This relationship may hold beyond Denmark, although comparative studies in other countries, as well as studies extending the scope to test type specific effects, seasonality, etc. would be required to ascertain this and could provide further insight.

Finally, we estimated the yearly municipal age-and sex-standardised case reporting, adjusted for the varying level of testing intensity, which we believe is a direct and better estimate of the underlying Legionnaires' disease incidence than the unadjusted estimate. We justified this based on the assumption that the diagnostic testing intensity for Legionnaires' disease depends on the overall community acquired pneumonia incidence, including by non-Legionella infections. We assume so due to the non-specificity of the Legionnaires' disease clinical presentation, the higher volume of *Legionella* testing and CAP compared to *Legionella* infections, and the observed similarity in seasonality between diagnostic *Legionella* testing and CAP. We believe that may cause, in the presence of underascertainment, an indirect effect of no interest in our surveillance data between location (municipality) and time (year) on disease reporting.

A strength of our study is that the incidences are calculated at municipal level with appropriate partial pooling through hierarchical spatio-temporal models to address instability of small area estimates due to randomness. Our estimates range from 1.4 to 12.0 per 100 000 inhabitants across municipalities. A number of studies have reported national incidences. Examples are 1.2 cases per 100,000 in USA in 2023 [30], 3.06 in Canada in 2015–2019 [18], 3.9 in New Zealand in 2010–2020 [65], and 5.6 in Switzerland in 2020, with the canton Ticino having reached up to 14.3 cases/100 000 [13,66]. The notification rates are strongly year dependent, with most studies explicitly reporting increasing rates in the last 20 years. Also, in the EU/EEA, the annual age-standardized notification rate has increased over the last decade, from 1.1 in 2013 to 2.7 in 2023, with significant intercountry variation: several countries registered under 0.5 cases/100,000, while Slovenia registered in 2019 a peak national rate of 8.4 / 100,000 [11].

Our Bayesian approach allowed us to estimate exceedance (and deficit) probabilities. This way, we identified municipalities and years with a high probability of excess Legionnaires' disease incidence, as well as units where very likely excess space-time interaction was at least partially indicative for increased environmental contamination or outbreak conditions. A recent study describes the advantages and superiority of Bayesian methods, including for the identification of outbreak areas [64]. Increasing the temporal resolution of the analysis, e.g., to monthly (and accounting for seasonality), could become the basis for future routine municipal-level incidence monitoring and outbreak detection or validation.

Our model identified Odense (Fyn) in 2015 as the only spot where the space-time interaction was with over 95% probability beyond the expected. This coincides with an outbreak of Knoxville ST9, as described by the public health authorities [67]. Kolding and Odense were the municipalities with high risk of Legionnaires' disease over several years, and merit further comparative study of conditions in the natural environment (e.g., microclimate, soil or waterreservoirs), built environment (e.g., prevalence of water utility structures and cooling towers), behaviour (e.g., use of airconditioning, misting

devices, spa pools or other man-made water systems), or bacterial strains endemic in the area (e.g., species, serogroup, MAb type, sequence type), possibly contrasting with municipalities where very low incidence are found.

Apart from Odense, also in Randers an increased case reporting had triggered attention of public health authorities in the past years. Randers is, as only area, clearly highlighted in the spatially unstructured effect (S16 Fig 19 in S1 Appendix). The area shows over 80% exceedance probability for the spatio-temporal interaction (Fig 7). The latter is also true for the following areas, mostly in one of the years: Koebenhavn, Frederiksberg, Gladsaxe, Roedovre, Hoersholm, Helsingoer, Aarhus, Norddjurs, Randers, Favrskov, Roskilde, Vejle, Aabenraa.

As the total case count included travel-associated cases with non-local exposure locations, we have tested the sensitivity of our results by excluding these. The exceedance probabilities for the relative case reporting and for the spatio-temporal interaction, after exclusion of travel-associated cases, are mapped in the Supplementary Information (S20 c Fig 25-27 in S1 Appendix). The results, e.g., the detection of Odense as cluster in 2015, remain consistent with our main conclusions.

Our study has also limitations, some of which relate to the data. The surveillance case definition of Legionnaires' disease aims to capture in-healthcare-tested disease. We can distinguish two non-diagnostic pathways. A proportion of legionella-caused pneumonia may not present at clinical care. However, in a country with good access to healthcare, pneumonia cases (contrary to legionellosis) not presenting at primary or secondary care are expected to be infrequent. Another proportion of legionella-caused pneumonia may present at clinical care but may not receive a Legionnaires' disease test. Our hierarchical model allows -in principle-, through partial pooling, to mutually learn between areas with higher testing and areas with lower testing, but this learning is not without limitations and compounding with measurement error due to sensitivity and specificity of the tests against different Legionella strains. Also related to data, we only used one data source to determine the location of cases, resulting in a small number of missing cases for the modeling. As these were evenly distributed, we do not believe this affects our conclusions.

Important limitations are related to the modelling assumptions. A first important caveat is that by adjusting for testing intensity, we may induce collider bias. Adjusting for an effect (Leg-tested) of a collider (Pneumonia), has the effect of partially adjusting for the collider itself, namely partially opening the biasing path (S1 Fig 2 in S1 Appendix). Simulation studies could be conceived to assess quantitatively the potential impact of such bias on the strength of our conclusions. We suggest that, awaiting such analyses, Danish public health authorities that wish to formulate public health action, compare our results with the total effect of the unadjusted model (S18 and 19 in S1 Appendix). We propose to be particularly cautious for those regions where this sensitivity analysis reveals a large discrepancy, considering that the unbiased direct effect is expected to lay between both model outputs (S20 b in S1 Appendix). We nevertheless believe that adjusting for the tested population is a necessary part of any advanced incidence estimation at population level. Our DAG also makes the strong assumption that there is no unmeasured confounding, e.g., by physician's awareness, being a common cause of both the *Legionella* diagnostic testing intensity and the Legionnaires' disease reporting (S1 Fig 3 in S1 Appendix). It is very plausible that there are such factors, for example if local or regional media coverage raises awareness among physicians, which will likely influence both the testing intensity and the reporting of cases.

Another potentially biasing factor to carefully consider in future studies is urbanization degree, which may be related both to exposure risk (e.g., density of cooling towers) and to the probability of diagnosis given being tested positive.

Furthermore, the acyclic nature of a DAG does not allow for plausible circular causation between case reporting and testing. Even in our suggested alternative DAG, with physician's awareness as unmeasured confounder, we do not capture that reporting of cases may trigger increased testing, followed by even higher case reporting. Although we choose here a static cross-sectional spatio-temporal modeling framework, we modeled the observed data of diagnosed and reported Legionnaires' disease explicitly recognizing the unobserved true disease incidence state.

To complement our study approach, and address the above drawbacks, we suggest that future studies employ dynamic state space models to estimate the evolving latent but hidden disease process, separately from the observation time

series. Future studies may also explore the use of measurement error models, to account for sensitivity and specificity of the tests, related to test type (e.g., PCR *L.pneumophila* vs *Legionella* spp.), sample quality (e.g., lower vs upper respiratory samples), and prevalent bacterial strains (e.g., species, serogroup, sequence types). Similarly, future models may distinguish settings of infection, being travel-associated, community-acquired, or hospital-acquired.

For countries without a centralized national laboratory database, to do a similar study as ours would require the identification of other data sources. Alternatively, well designed probability sampling surveys from hospitals and their clinical laboratories may be performed. This would come with additional modelling challenges and corresponding assumptions for interpolation towards areal data. However, although the results obtained are specific to the Danish context, the generic data requirements and modelling assumptions are expected to be similar for a different context.

## Conclusions

Despite limitations and possible bias, our study of testing-adjusted case reporting suggests that no substantial increase in Legionnaires' disease has occurred during the nine-year study period. Instead, the case ascertainment by physicians has improved considerably through increased *Legionella* testing, particularly among elderly patients.

Our assessment of the Legionnaires' disease testing intensity in Denmark shows major spatio-temporal variation across the country. We also quantified the positive effect of testing intensity on Legionnaires' disease reporting and found a threshold of annually testing slightly over 1% of the population above which the yield of new cases does not further improve. Studies in other countries would be useful to assess the generalizability of these findings. Finally, we obtained testing-adjusted estimates of the spatio-temporal variation of Legionnaires' disease over the Danish territory.

Our estimates and their uncertainty, given the model and the data, give physicians a potential benchmark about the diagnostic behavior at municipality level. Furthermore, public health authorities could differentiate and target diagnostic guidance, taking into consideration the actual diagnostic testing intensity and case detection. The authorities can also better direct resources for environmental research of areas with exceptionally high or low testing-adjusted disease incidence. Therefore, we believe our enhanced surveillance study results give important clues to improve diagnostic testing guidance and research into risk factors of Legionnaires' disease, and ultimately to improve Legionnaires' disease prevention and control.

## Supporting information

**S1 Appendix. Contains contextual data, extended methods and model information, and supplementary results.** (PDF)

## Acknowledgments

We are grateful to the Departments of Clinical Microbiology, Denmark, for their contribution to the national surveillance and the Danish Microbiology Database project (MiBa), and the staff of Statens Serum Institut who curate the MiBa register data. Our gratitude goes also to the clinicians who notified cases of Legionnaires' disease to the Statens Serum Institut. We thank Miguel Hernán for the helpful discussion on the causal diagram and Olivier Cecchi for support with querying OpenStreetMap.

**\*\*Disclaimer**: The views and opinions expressed herein are the authors' own and do not necessarily state or reflect those of ECDC. ECDC is not responsible for the data and information collation and analysis and cannot be held liable for conclusions or opinions drawn.

## Author contributions

**Conceptualization:** Emmanuel Robesyn, Søren Anker Uldum, Charlotte Kjelsø.

**Data curation:** Emmanuel Robesyn, Karsten Dalsgaard Bjerre.

**Formal analysis:** Emmanuel Robesyn, Christel Faes.

**Methodology:** Emmanuel Robesyn, Christel Faes.

**Supervision:** Marc Struelens, Cecilia Stålsby Lundborg, Steen Ethelberg, Christel Faes.

**Validation:** Søren Anker Uldum, Karsten Dalsgaard Bjerre, Charlotte Kjelsø.

**Visualization:** Emmanuel Robesyn.

**Writing – original draft:** Emmanuel Robesyn.

**Writing – review & editing:** Søren Anker Uldum, Karsten Dalsgaard Bjerre, Charlotte Kjelsø, Marc Struelens, Cecilia Stålsby Lundborg, Steen Ethelberg, Christel Faes.

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
