## [Decision Letter · Decision Letter 0]

13 Oct 2025

Dear Dr. Robesyn,

We look forward to receiving your revised manuscript.

Kind regards,

Marta Palusinska-Szysz

Academic Editor

PLOS ONE

Journal Requirements:

2. We note that Figures 2, 4, 5, 6, and 7 in your submission contain map images which may be copyrighted. All PLOS content is published under the Creative Commons Attribution License (CC BY 4.0), which means that the manuscript, images, and Supporting Information files will be freely available online, and any third party is permitted to access, download, copy, distribute, and use these materials in any way, even commercially, with proper attribution. For these reasons, we cannot publish previously copyrighted maps or satellite images created using proprietary data, such as Google software (Google Maps, Street View, and Earth). For more information, see our copyright guidelines: http://journals.plos.org/plosone/s/licenses-and-copyright.

a. You may seek permission from the original copyright holder of Figures 2, 4, 5, 6, and 7 to publish the content specifically under the CC BY 4.0 license.

3. Please ensure that you refer to Figures 5, 6, and 7 in your text as, if accepted, production will need this reference to link the reader to the figures.

4. Please upload a copy of Figures 8 and 9, to which you refer in your text on pages 19 and 20. If the figure is no longer to be included as part of the submission please remove all reference to it within the text.

5. Please remove all personal information, ensure that the data shared are in accordance with participant consent, and re-upload a fully anonymized data set.

**Comments to the Author**

1. Is the manuscript technically sound, and do the data support the conclusions?

Reviewer #1: Partly

2. Has the statistical analysis been performed appropriately and rigorously?

Reviewer #1: Yes

3. Have the authors made all data underlying the findings in their manuscript fully available?

Reviewer #1: No

4. Is the manuscript presented in an intelligible fashion and written in standard English?

Reviewer #1: Yes

Reviewer #1: Robesyn et al. present an interesting paper considering heterogeneity in LD testing and case reporting in space/time in Denmark. The analysis is of interest and benefits from multiple national-level data sources. I have the below comments which I hope will strengthen the manuscript further:

Abstract:

- Please could you briefly define (perhaps in brackets after ‘Legionella Test’) what legionella tests you are including (i.e UAT and PCRs)

- ‘Conform a positive time effect’ does not seem a grammatically correct ending to this sentence - please improve

Introduction:

- Typo line 44: ‘unknown to WHAT extent’ not ‘which’ extent

- Reformulation proposal lines 45-48:

‘Despite the global occurrence [2], the clinical severity [3], the public health burden (mostly attributable to years of life lost [4, 5]), and the treatable and preventable

character of this waterborne respiratory disease, many low- and middle-income countries

have no effective surveillance in place.’

- Line 48: add ‘within’ after including

Methods:

- Lines 137-138 ‘Latent Legionnaires’ disease incidence’. We would not typically refer to LD as being ‘latent’ as we would for other communicable diseases like TB. I think ‘undiagnosed’ may be more appropriate for the presumed meaning behind the sentence. Please amend here and elsewhere in the manuscript where LD is referred to as latent.

- Have case definitions remained consistent throughout the study period? Are they amended in the context of LD outbreaks?

- Lines 143-144 should be ‘both’ not ‘either’.

Results:

- Were there changes in the proportion of probable/confirmed cases over time?

- Line 303-304 please can you explain the sentence 83% of all positive tested persons in the database. Does this mean that the reported number (2183) is only 83% of the number of cases identified via the MiBa? I thought in lines 167-169 you stated the difference between the number of cases from both sources were negligible?

- To aid understanding and readability I think a more formal definition of RRT would be helpful. What is it relative to? i.e. what number would give an RRT of 1? I have the same point for RRC.

- Figure 3(b) the graph appears to have a more natural transition point somewhere between 1100-1200 tested persons/100,000. What is the rationale behind identifying 1000 as the transition point? That appears to still be on the upward slant of the curve. I think this graph would benefit from being made significantly larger - my understanding is that this is one of the most significant findings in the paper.

- Line 369 has ‘case case’

Discussion:

- The continued change in UAT and PCR testing over time is very interesting and deserves more comment/explanation in the discussion.

- When describing the diagnostic testing intensity I think there should be some acknowledgement that this will depend to a certain degree on the underlying ‘true’ incidence of disease over time (i.e. if there are more cases in the population, then the curve of testing intensity against case reporting may appear different). An acknowledgement that this may differ in different times, in addition to different places, would be helpful.

- Line 542 typo: areal

Conclusion:

- The conclusion ‘no substantial increase in true LD has occurred during the nine-year study period’ is too strong given the described limitations in the discussion, and is discordant with how the findings are more cautiously presented throughout the rest of the manuscript. Suggest amending to ‘suggests that’ or ‘evidence in favour of’.

.

Reviewer #1: No

---

## [Author Response · Author response to Decision Letter 1]

3 Dec 2025

We would like to thank you and the Reviewer for the valuable comments.

We submit a revised version of our manuscript, taking into account the points raised during the review process. We have carefully addressed the comments and reply below to each of them, clarifying also what was adapted in the manuscript.

Comments by Academic Editor:

We have edited the manuscript according to PLOS ONE’s style requirements. Files were renamed as required. We numbered the tables and figures and uploaded them separately. Textual changes are reflected in the revision file labelled 'Revised Manuscript with Track Changes'.

2. We note that Figures 2, 4, 5, 6, and 7 in your submission contain map images which may be copyrighted. All PLOS content is published under the Creative Commons Attribution License (CC BY 4.0), which means that the manuscript, images, and Supporting Information files will be freely available online, and any third party is permitted to access, download, copy, distribute, and use these materials in any way, even commercially, with proper attribution. For these reasons, we cannot publish previously copyrighted maps or satellite images created using proprietary data.

We have replaced the basemap of the Danish municipalities with data adapted from OpenStreetMap, available under the Open Database License, and we have added this information in the map legend. Furthermore, we have inserted the following information in the Material and Methods section: “The basemap of the Danish municipalities was adapted from OpenStreetMap, available under the Open Database License (openstreetmap.org/copyright).”

3. Please ensure that you refer to Figures 5, 6, and 7 in your text as, if accepted, production will need this reference to link the reader to the figures.

We have inserted reference to Figure 5. We have replaced the reference to Figure 8 and 9 by Figure 6 and 7, respectively.

4. Please upload a copy of Figures 8 and 9, to which you refer in your text on pages 19 and 20. If the figure is no longer to be included as part of the submission please remove all reference to it within the text.

We have replaced the reference to Figures 8 and 9 by Figure 6 and 7, respectively.

5. Please remove all personal information, ensure that the data shared are in accordance with participant consent, and re-upload a fully anonymized data set.

We have reviewed the data shared through the manuscript, the supplementary information, and the repositories, considering the Data Policy and the guidance in the recommended article. The data provided are anonymized.

We provide observed data counts in Table 1 and 2 in the manuscript and Tables 1-4 in the Supplementary Information S4-S6. Further, in the Supplementary Information S21-S24 we provide extensive look-up tables with the exact values of the estimates as shown in the maps of the manuscript and further results from sensitivity analyses. Demographic and map data, along with computing code, are available at the GitHub and Zenodo repository. For access to additional data, we explain in the data availability statement how these can be requested.

6. If the reviewer comments include a recommendation to cite specific previously published works, please review and evaluate these publications to determine whether they are relevant and should be cited.

The reviewer did not include a recommendation to cite specific previously published works.

Comments by Reviewer #1

Robesyn et al. present an interesting paper considering heterogeneity in LD testing and case reporting in space/time in Denmark. The analysis is of interest and benefits from multiple national-level data sources. I have the below comments which I hope will strengthen the manuscript further:

We are grateful for the comments and acknowledge that addressing them has significantly improved the manuscript.

Abstract:

- Please could you briefly define (perhaps in brackets after ‘Legionella Test’) what legionella tests you are including (i.e. UAT and PCRs).

Thank you for highlighting this important aspect. We have inserted the test types as follows: “persons receiving at least one Legionella urinary antigen or PCR test per 100,000 inhabitants.”

- ‘Conform a positive time effect’ does not seem a grammatically correct ending to this sentence – please improve.

We have replaced this by “reflecting an upward trend.”

Introduction:

- Typo line 44: ‘unknown to WHAT extent’ not ‘which’ extent

We have replaced the word, both in the Introduction and in the Abstract.

- Reformulation proposal lines 45-48: ‘Despite the global occurrence [2], the clinical severity [3], the public health burden (mostly attributable to years of life lost [4, 5]), and the treatable and preventable character of this waterborne respiratory disease, many low- and middle-income countries have no effective surveillance in place.’

We agree that the suggested reformulation improves the readability and have adopted the proposal.

- Line 48: add ‘within’ after including

We have inserted the word.

Methods:

- Lines 137-138 ‘Latent Legionnaires’ disease incidence’. We would not typically refer to LD as being ‘latent’ as we would for other communicable diseases like TB. I think ‘undiagnosed’ may be more appropriate for the presumed meaning behind the sentence. Please amend here and elsewhere in the manuscript where LD is referred to as latent.

Indeed, the meaning of ‘latent’ in our manuscript does not refer to the infection status of an individual. Instead, it refers to the unmeasured or hidden disease generating process or true disease incidence in the population, which is our surveillance target of interest that we aim to estimate. To avoid confusion, we have replaced the word ‘latent’ by ‘true’ Legionnaires disease incidence. Similarly, we have also adapted the formulation of latent state in the

Discussion.

- Have case definitions remained consistent throughout the study period? Are they amended in the context of LD outbreaks?

The case definition of Legionnaires’ disease used in Denmark, including in the context of outbreaks, is based on the official EU surveillance case definition. Outbreak case definitions differ mainly by restriction in time, place and person, but not in their clinical and laboratory criteria. Throughout the study period there was one change in the EU case definition, namely in 2018. The 2002 definition, valid until 25/07/2018, defined both Legionnaires’ disease and Pontiac fever, but this has not affected our study data, which is focussed on Legionnaires’ disease only. In both versions, a positive urinary antigen test was considered a confirmatory case criterium, while a positive PCR test classified the case as probable. This distinction in case classification was not deemed relevant to our study and was therefore not used. We note here that a large proportion (around 30-40%) of the PCR-positive cases in Denmark are confirmed by culture.

We have inserted the 2002 definition as reference in the Introduction of our manuscript, for completeness. We added a clarification in the Methods section: “Cases fulfilled the EU case definition of confirmed or probable case. Throughout the study period, the EU case definition considered a positive urinary antigen test as a confirmed case criterium, and a positive PCR test as a probable case criterium, but this distinction in case classification was not deemed relevant to our study and was therefore not used.”

- Lines 143-144 should be ‘both’ not ‘either’.

Indeed, both may influence the reported case count. We have replaced the word.

Results:

- Were there changes in the proportion of probable/confirmed cases over time?

For this study, we did not collect the variable ‘case classification’ but we can provide relevant background. Per the case definition in use during the study period, the proportion of probable and confirmed cases is largely (but not only) defined by the test types used: a positive urinary antigen test or a culture with isolation of Legionella spp. contributes to the case classification ‘confirmed’, while a positive PCR test contributes to a case being labelled as ‘probable’. As the proportion of urinary antigen test decreased and the proportion of PCR increased over the study period, the proportion of ‘confirmed’ cases has decreased. However, the binary surveillance classification ‘confirmed/probable case’ does not capture sufficiently the complexity of the test sensitivity and specificity of the different tests for the variety of Legionella species and serogroups causing clinical Legionnaires’ disease. We describe the most salient changes in use of test types over time in the paragraph before table 1, and we refer for more detail to Table 3 in the Supplementary Information S6 where we report on the within-year test profiles of the tested persons.

- Line 303-304 please can you explain the sentence 83% of all positive tested persons in the database. Does this mean that the reported number (2183) is only 83% of the number of cases identified via the MiBa? I thought in lines 167-169 you stated the difference between the number of cases from both sources were negligible?

Thank you for the question. To explain the difference, we refer to three nodes in Figure 1: ‘Pos Leg-tested’ or people who have tested positive for a Legionella test, ‘Diagnosed LD’ or people who were found to have Legionnaires’ disease by their clinician, and ‘Reported LD’ or people who’s diagnosis was recorded in the surveillance database by the Department of Infection Epidemiology and Prevention, Statens Serum Institut.

There is a significant difference between the number of positive Legionella tested persons and diagnosed cases (for which the reasons are explained in the manuscript), but a negligible difference between the number of diagnosed cases and reported cases.

We understand that our statement about ‘automated electronic reporting of data from the national microbiological database’ was misleading and did not capture that there is still a verification of the diagnosis with the clinician. The automated ‘detection algorithm’ of positive tests on the national microbiological database does not lead to automated ‘case reporting’ in the surveillance system but to verification of the diagnosis with the clinician.

We have corrected our manuscript text: “However, this is assumed negligible in Denmark since 2014 due to an automated detection algorithm on the national microbiological database, whereby for each positive test, the diagnosis is verified with the clinician.”

- “To aid understanding and readability, I think a more formal definition of RRT would be helpful. What is it relative to? i.e. what number would give an RRT of 1? I have the same point for RRC.”

We agree that the link between the model equations in the Methods and the terminology in the Results section was not explicit enough. The value of one for the relative risk (RR) refers to testing intensity or case reporting as expected based on the intensity or reporting in the whole study area. We have now inserted in the Methods section a clarifying text, with more formal definitions:

“In the Results section, we call the relative risk from the two models, respectively, the ‘relative testing intensity (RRT)’ and the ‘relative case reporting (RRC)’, i.e. the multiplicative factor for a specific year and municipality as compared to the expected value. A value of two means that the testing intensity or case reporting was twice as large as expected from the overall intensity/reporting in the whole country.”

- Figure 3(b) the graph appears to have a more natural transition point somewhere between 1100-1200 tested persons/100,000. What is the rationale behind identifying 1000 as the transition point? That appears to still be on the upward slant of the curve.

We have been hesitant to make strong conclusions about the trend above a testing intensity above 1000 tested persons/100,000 taking into consideration the slightly lower point estimates beyond 1200 and the larger uncertainty interval (Fig 3, b and S14 Fig 17). We have edited the text to capture better the continued increase of the point estimate up to 1200:

“Further, we estimated that with increasing testing intensity, the relative case reporting (on log scale) increased (Fig 3, b and S14 Fig 17). The increasing relative case reporting is observed up to about 1000 to 1200 tested persons per 100 000 inhabitants, with relative case reporting of about 1.6 (log(RRC) = 0.5). At higher testing intensities, there is no further improvement in case reporting and there is larger uncertainty of the estimate.”

- I think this graph would benefit from being made significantly larger - my understanding is that this is one of the most significant findings in the paper.

We agree that Fig 3, b is an essential finding and we have increased its size. At this occasion, we have also made the caption of Fig 3 more accurate.

- Line 369 has ‘case case’

Thank you for noting.

Discussion:

- The continued change in UAT and PCR testing over time is very interesting and deserves more comment/explanation in the discussion.

We fully agree and have elaborated on this point in the Discussion as follows:

“Whereas the population of Denmark between 2014 and 2022 has increased with 4 %, the number of persons tested for Legionella in a given year has increased with 141%. While diagnostic testing intensity may not be known, a similar pattern may take place in other countries. Many reports and studies signalling apparent increased Legionnaires’ disease incidence without considering the testing intensity will therefore not reliably reflect true Legionnaires’ disease incidence.

Where since the beginning of the century urinary antigen test availability has drastically changed the diagnostic landscape, we described during our study period a shift from urinary antigen test use to PCR use, further boosted by PCR lab capacity created during the COVID-19 pandemic. We considered in our study the total testing intensity and did not focus on the test characteristics of the urinary antigen and PCR tests, but these are also key to understand the yield of the tests and thus the resulting reporting incidence.”

- When describing the diagnostic testing intensity I think there should be some acknowledgement that this will depend to a certain degree on the underlying ‘true’ incidence of disease over time (i.e. if there are more cases in the population, then the curve of testing intensity against case reporting may appear different). An acknowledgement that this may differ in different times, in addition to different places, would be helpful.

Indeed, in line with our DAG in Fig 1, the true disease will affect the testing intensity over time. We have reinforced the importance of the time dimension, in parallel to the spatial dimension, in the following way. Where we wrote “Even in our suggested alternative DAG, with physician’s awareness as unmeasured confounder, we do not capture that reporting of cases may trigger increased testing, followed by even higher case reporting.”, we have added “Such short as well as long changes over time may happen simultaneous to changes in true disease incidence, e.g. due to weather and climatic conditions, clearly making the inference of true disease incidence a major challenge.”

- Line 542 typo: areal

We believe the spelling is correct, referring to an area. Our modelling was based on counts in (administrative) area’s. When data were to come from hospitals, these counts would need to be attributed to a recruitment area to know the denominator for incidence calculation and modelling. We edited the sentence slightly to improve the structure.

Conclusion:

- The conclusion ‘no substantial increase in true LD has occurred during the nine-year study period’ is too strong given the described limitations in the discussion, and is discordant with how the findings are more cautiously presented throughout the rest of the manuscript. Suggest amending to ‘suggests that’ or ‘evidence in favour of’.

Thank yo

---

## [Decision Letter · Decision Letter 1]

2 Feb 2026

Diagnostic testing intensity for Legionnaires’ disease: spatio-temporal assessment and its effect on surveillance case reporting, Denmark, 2014-2022

PLOS One

Dear Dr.  Robesyn,

Thank you for submitting your manuscript to PLOS ONE. After careful consideration, we feel that it has merit but does not fully meet PLOS ONE’s publication criteria as it currently stands. Therefore, we invite you to submit a revised version of the manuscript that addresses the points raised during the review process.

We look forward to receiving your revised manuscript.

Kind regards,

Marta Palusinska-Szysz

Academic Editor

PLOS One

Journal Requirements:

Reviewer's Responses to Questions

**Comments to the Author**

Reviewer #2: (No Response)

2. Is the manuscript technically sound, and do the data support the conclusions?

Reviewer #2: Yes

3. Has the statistical analysis been performed appropriately and rigorously?

Reviewer #2: Yes

4. Have the authors made all data underlying the findings in their manuscript fully available?

Reviewer #2: Yes

5. Is the manuscript presented in an intelligible fashion and written in standard English?

Reviewer #2: Yes

Reviewer #2: The study provides timely and important results for developing surveillance approaches to LD. The analysis uses an extensive national dataset of LD cases and diagnostic testing in Denmark. The modelling approach is novel and relevant to the field, the manuscript is generally well presented, and the figures and tables are clear. My comments mainly concern the robustness of the conclusions.

Major comments

• Throughout the manuscript, especially in the abstract and conclusions, the interpretation shifts from testing-adjusted reporting (as a proxy for underlying incidence) to statements about “true” LD incidence. Even with the cautious wording (“suggest”), this may be read as a claim about the underlying disease process rather than the surveillance process. The analysis does not account for all uncertainties in diagnostic/ascertainment (e.g., it includes only cases that enter the diagnostic pathway), and the manuscript notes that adjusting for testing intensity can introduce collider bias. A revision of the abstract and conclusions to focus on testing-adjusted reporting and to reflect these limitations will avoid misinterpretation.

• The manuscript interprets spatially structured and space–time interaction effects as municipality-level clustering and potentially local environmental conditions. However, travel-associated cases, assigned to the municipality of residence rather than the likely exposure location, are included. Some spatial patterns may therefore reflect non-local exposures, affecting how “local” risk patterns are interpreted. Clarifying the rationale for including travel-associated cases, or providing a sensitivity analysis excluding them, would make the conclusions more reliable.

• A relevant potential confounder that is not clearly addressed is urbanicity. Urbanicity may relate to both i) testing intensity and ii) exposure opportunities (e.g., higher density of cooling towers in urban areas). This could bias the estimated testing–case relationship and the interpretation of testing-adjusted spatial patterns.

Minor comments

• L116: The prevention of LD depends not only on diagnostic guidelines and surveillance, but also on effective risk management of pathogenic Legionella species in engineered water systems. Please revise to reflect this broader prevention context.

• L257: The manuscript describes the testing effect as linear, while the SI indicates the results come from a nonlinear RW1 smooth over binned testing intensity. Please clarify the model specification in the main text to avoid confusion.

• Figure 3 presents results on the logarithmic scale. For most readers, would it be easie to interpret on the arithmetic scale?

.

Reviewer #2: **Yes:** Émile SylvestreÉmile SylvestreÉmile SylvestreÉmile Sylvestre

---

## [Author Response · Author response to Decision Letter 2]

14 Mar 2026

We thank the second reviewer for the highly relevant comments. We believe that addressing them has significantly improved our manuscript.

Please note that the attached document 'Response to reviewers_2' contains the figures pertaining to this reponse.

Reviewer:

The study provides timely and important results for developing surveillance approaches to LD. The analysis uses an extensive national dataset of LD cases and diagnostic testing in Denmark. The modelling approach is novel and relevant to the field, the manuscript is generally well presented, and the figures and tables are clear. My comments mainly concern the robustness of the conclusions.

Major comments

1. Throughout the manuscript, especially in the abstract and conclusions, the interpretation shifts from testing-adjusted reporting (as a proxy for underlying incidence) to statements about “true” LD incidence. Even with the cautious wording (“suggest”), this may be read as a claim about the underlying disease process rather than the surveillance process. The analysis does not account for all uncertainties in diagnostic/ascertainment (e.g., it includes only cases that enter the diagnostic pathway), and the manuscript notes that adjusting for testing intensity can introduce collider bias. A revision of the abstract and conclusions to focus on testing-adjusted reporting and to reflect these limitations will avoid misinterpretation.

We thank the reviewer for the thoughtful comment. We agree that the testing-adjusted reporting, our analysis outcome, is an imperfect proxy for the underlying incidence. Aspects like cases not presenting to healthcare, measurement error in the tests, dynamic interactions, unmeasured confounders, etc. are not accounted for, and we cannot exclude partial (i.e. depending on the strength of association between collider and descendant) collider bias. To avoid misinterpretation or overconfidence, we have removed ‘true’ where we refer to our work and meant latent or underlying incidence and adapted the wording in the abstract, the main text (original lines 127, 139, 148, 483, 579) and the conclusions.

Nevertheless, we do not want to obfuscate that one estimand of our surveillance study is indeed the underlying disease incidence, which -through our proxy- we tried to estimate to the best of our ability. We therefore feel it is needed to cautiously conclude on the underlying incidence or disease process, without suggesting it is ‘true’. The reasoning is similar to the argument that a causal analysis should, despite limitations, conclude on causal effects and not limit its conclusions to merely describing associations (Hernan, 2018). However, we have carefully avoided to refer to the underlying incidence or process in the Results section, and we refer in that section only to ‘testing-adjusted reporting’.

Reference: Hernán MA. Am J Public Health. 2018 May;108(5):616-619. doi: 10.2105/AJPH.2018.304337. Epub 2018 Mar 22. URL: https://pubmed.ncbi.nlm.nih.gov/29565659/

We are grateful for pointing at the limitation of the non-diagnostic path and we have inserted this aspect at the start of our limitations section. We take the opportunity to elaborate here a little further.

The surveillance case definition of Legionnaires’ disease indeed only aims to capture in-healthcare-tested disease. A proportion of legionella-caused pneumonia may not present at clinical care. However, in a country with good access to healthcare, pneumonia cases (contrary to legionellosis) not presenting at primary or secondary care are expected to be infrequent.

Another non-diagnostic pathway would be legionella-caused pneumonia presenting at healthcare but not receiving a Legionnaires’ disease test. Our hierarchical model allows -in principle-, through partial pooling, to mutually learn between areas with higher testing and areas with lower testing. This learning is not without limitations and future (simulation) studies may assess whether post-stratification, e.g. simulating a situation where all areas have very high testing, would provide better incidence estimates. Such studies could include measurement error models where test characteristics like sensitivity and specificity against different Legionella strains are taken into account.

In countries with generalised under-ascertainment, i.e. low testing in all or most areas, learning through partial pooling would be poor, and the non-diagnostic pathway would be an even more important limitation.

2. The manuscript interprets spatially structured and space–time interaction effects as municipality-level clustering and potentially local environmental conditions. However, travel-associated cases, assigned to the municipality of residence rather than the likely exposure location, are included. Some spatial patterns may therefore reflect non-local exposures, affecting how “local” risk patterns are interpreted. Clarifying the rationale for including travel-associated cases, or providing a sensitivity analysis excluding them, would make the conclusions more reliable.

Our primary estimand was the effect of testing intensity on the total Legionnaires’ disease reporting, the main indicator in many surveillance reports, and for that reason travel-associated cases were included. However, inferring local environmental risk from total case incidence may indeed be biased by the inclusion of travel-associated cases. To test the sensitivity of our results to this inclusion, we have now also made estimates excluding travel-associated cases and we have added this sensitivity analysis to the supplementary material. The estimated case reporting and the exceedance probabilities for both the relative case reporting and the spatio-temporal interaction are shown in S20 Fig 25-27, and at the end of this document. The results, e.g. the detection of Odense as cluster in 2015, remain consistent with our main conclusions. We have added this information also to the Discussion section.

3. A relevant potential confounder that is not clearly addressed is urbanicity. Urbanicity may relate to both i) testing intensity and ii) exposure opportunities (e.g., higher density of cooling towers in urban areas). This could bias the estimated testing–case relationship and the interpretation of testing-adjusted spatial patterns.

We agree that it is important to assess the potential biasing effect of urbanization degree on our estimates. Doing so on https://dagitty.net/, we found that adjustment for urbanization degree is only needed if we make the additional assumption that urbanization is related to the probability of being diagnosed, given being tested positive. Note that in the below graphs the arrow between Legionella infections and pneumonia is removed for illustrative purpose as in Figure 3 of Supplement S1, to highlight the relevant causal paths.

In the first graph below, where urbanization is influencing the presence of Legionella determinants (e.g. density of cooling towers) as well as the testing intensity (e.g. higher in cities), the path is blocked on Legionella determinants, whether or not we adjust for Legionella testing intensity.

[Figure in attached document]

In the alternative graph below, where the area determines the degree of urbanization, and the latter influences the testing intensity, we must adjust for urbanization to measure the direct path from municipality to reported cases (a). However, that is not needed anymore once we adjust for testing intensity (Leg-tested as white circle) (b).

[Figures in attached document]

Finally, when we consider that urbanization degree not only relates to testing intensity but also to the probability of being diagnosed, given being tested positive, then indeed bias is possible and adjustment for urbanization degree is needed.

[Figure in attached document]

We believe this may be a realistic assumption. At the time of our study this data was not available. We have mentioned this relevant factor in the Discussion, for consideration in future studies.

Minor comments

4. L116: The prevention of LD depends not only on diagnostic guidelines and surveillance, but also on effective risk management of pathogenic Legionella species in engineered water systems. Please revise to reflect this broader prevention context.

Thank you for the remark. We agree and have inserted this critical preventive element to the text.

5. L257: The manuscript describes the testing effect as linear, while the SI indicates the results come from a nonlinear RW1 smooth over binned testing intensity. Please clarify the model specification in the main text to avoid confusion.

Thank you for flagging this. We have corrected the notation and accompanying text.

6. Figure 3 presents results on the logarithmic scale. For most readers, would it be easier to interpret on the arithmetic scale?

Indeed, this will help the interpretation of our results. We have now plotted the relative testing intensity and the relative case reporting on the arithmetic scale.

[Figure in attached document]

Three Figures added to supplement S20, sensitivity analyses: relative case reporting and exceedance probabilities from data excluding travel-associated cases.

[Figures in attached document]

---

## [Editor Report · Decision Letter 2]

22 Mar 2026

Diagnostic testing intensity for Legionnaires’ disease: spatio-temporal assessment and its effect on surveillance case reporting, Denmark, 2014-2022

PONE-D-25-22931R2

Dear Dr. Emmanuel Robesyn,

We’re pleased to inform you that your manuscript has been judged scientifically suitable for publication and will be formally accepted for publication once it meets all outstanding technical requirements.

Kind regards,

Marta Palusinska-Szysz

Academic Editor

PLOS One
---

## [Editor Report · Acceptance letter]

PONE-D-25-22931R2

PLOS One

Dear Dr. Robesyn,

I'm pleased to inform you that your manuscript has been deemed suitable for publication in PLOS One. Congratulations! Your manuscript is now being handed over to our production team.

Kind regards,

on behalf of

Dr. Marta Palusinska-Szysz

Academic Editor

PLOS One